# Evaluation of the wind farm parameterization in the Weather Research and Forecasting model (version 3.8.1) with meteorological and turbine power data

Joseph C. Y. Lee[1,2] and Julie K. Lundquist[1,2]

[1]Department of Atmospheric and Oceanic Sciences, University of Colorado, UCB 311, Boulder, CO 80309, USA
[2]National Renewable Energy Laboratory, Golden, CO, USA

*Correspondence to*: Joseph C. Y. Lee (chle6805@coloardo.edu)

**Abstract.** Forecasts of wind-power production are necessary to facilitate the integration of wind energy into power grids, and these forecasts should incorporate the impact of wind-turbine wakes. This paper focuses on a case study of four diurnal cycles with significant power production, and assesses the skill of the wind farm parameterization (WFP) distributed with the Weather Research and Forecasting (WRF) model version 3.8.1, as well as its sensitivity to model configuration. After validating the simulated ambient flow with observations, we quantify the value of the WFP as it accounts for wake impacts on power production of downwind turbines. We also illustrate with statistical significance that a vertical grid with nominally 12-m vertical resolution is necessary for reproducing the observed power production. Further, the WFP overestimates wake effects and hence underestimates downwind power production during high wind-speed, highly stable and low turbulence conditions. We also find the WFP performance is independent of the number of wind turbines per model grid cell and the upwind-downwind position of turbines. Rather, the ability of the WFP to predict power production is most dependent on the skill of the WRF model in simulating the ambient wind speed.

## 1 Introduction

In recent years, numerical weather prediction (NWP) models have become an indispensable tool in the wind-energy industry, not only in day-to-day wind-energy production forecasts (Wilczak et al., 2015), but also to support wide-scale wind-power penetration (Marquis et al., 2011) and wind resource assessment. To forecast power production accurately at wind farms, the simulation tools should resolve all physical processes relevant to the wind field, including possible impacts of the wind turbines themselves. Consequently, including the meteorological effects of wind farms in NWP models can improve power-production forecasts.

Researchers have developed various methods to numerically represent wind farms. Via large-eddy simulations (LES), some investigators assess the meteorological impacts of wind turbines as well as power production (Abkar and Porté-Agel, 2015b; Aitken et al., 2014; Calaf et al., 2010; Churchfield et al., 2012; Jimenez et al., 2007; Mirocha et al., 2014; Na et al.,

2016; Sharma et al., 2016; Wu and Porté-Agel, 2011). Simulating wind turbines and their effects in LES is, while useful,
computationally expensive, making wind-farm-scale simulations unreasonable in an operational setting.

At coarser spatial scales, suitable for global, synoptic or mesoscale models, numerically representing wind turbine effects may involve unrealistic assumptions. For example, researchers have used exaggerated surface roughness to represent the reduction of wind speed (WS) caused by wind farms in a global model (Barrie and Kirk-Davidoff, 2010; Frandsen et al., 2009; Keith et al., 2004). Similarly, the analytical wind park model of Emeis and Frandsen (1993) considers both the
downward momentum flux and the momentum loss due to surface roughness. The revised model by Emeis (2010) accounts for the spatially-averaged momentum-extraction coefficient by turbines, and the parameters become atmospheric-stability dependent. However, these models omit the consideration of turbine-scale interactions between the hub and the surface (Abkar and Porté-Agel, 2015a; Fitch et al., 2012, 2013b).

Aside from indirectly representing wind turbines via exaggerated roughness, another common approach is to use the
turbine power curve to deduce elevated drag and turbulence production of wind turbines. A power curve illustrates the relationship between inflow WS at hub height and power production of a particular turbine model. This method can model meteorological impacts of wind turbines and the impact of turbine drag force (Baidya Roy, 2011; Blahak et al., 2010). Based on this technique, Fitch et al. (2012) added the consideration of the turbine thrust coefficient to simulate both turbine drag and power loss.

In the wind farm parameterization (WFP) of the Weather Research and Forecasting (WRF) model, wind turbines in each model grid cell are collectively represented as a turbulence source and a momentum sink within the vertical levels of the turbine rotor disk (Fitch et al., 2012). A fraction of the kinetic energy extracted by the virtual wind turbines is converted to power, and the turbulence generation is derived from the difference between the thrust and power coefficients. In the WFP scheme, the use of the WS-dependent thrust coefficients accounts for the effects of local wind drag on wind-energy
extraction as well as on power estimation. The WRF WFP offers flexibility, where users can modify the parameters of a turbine model, such as its hub height, rotor diameter, power curve, and thrust coefficients, and does not require other empirically-derived parameters. By simulating wind farms in a mesoscale weather model, WRF users can simulate aggregated effects of wind-turbine wakes and thus the effects of power production of downwind turbines.

An approach similar to the WRF WFP proposed by Abkar and Porté-Agel (2015a) relies on an extra parameter, which is
the ratio of the freestream velocity to the horizontally-averaged hub-height velocity of a turbine-containing grid cell. This ratio depends on various factors such as the wind-farm density and layout, and requires preliminary simulation results (Abkar and Porté-Agel, 2015a). Therefore, the publicly-available WFP in the WRF model is chosen in this project for observed power comparison. On the other hand, the explicit wake parameterization (EWP) recently designed by Volker et al. (2015) uses classical wake theory to describe the unresolved wake expansion. Both the WRF WFP and the EWP average the
drag force within grid cells. Nevertheless, users of the EWP need to adjust the length scales that determine wake expansion in the EWP for different situations.

In this paper, we evaluate the WFP in the WRF model via comparison to actual power-production data. The WRF WFP has been widely used to assess the impacts of onshore and offshore wind farms at different spatial scales and in different stability regimes (Eriksson et al., 2015; Fitch et al., 2013a, 2013b; Jiménez et al., 2015; Lee and Lundquist, 2017; Miller et al., 2015; Vanderwende et al., 2016; Vanderwende and Lundquist, 2016; Vautard et al., 2014). Whereas WFP predictions have been compared to power production of offshore wind farms for a limited set of WS (Jiménez et al., 2015), here we explore a range of WS, wind direction (WD), turbulence, and atmospheric stability conditions. The large range of wind conditions induces spatially- and temporally-diverse power production, thereby providing a basis for a comprehensive evaluation of the WFP. The uniqueness of this project lies in the in-depth assessment of the WRF WFP performance in forecasting and simulating wind energy of a sizable onshore wind farm, using observed power-production data.

We describe the observation data and the model design in Section 2. In Section 3, we evaluate the simulations by comparing the meteorological and power-generation data with a statistical examination. We close with a proposal of improvements on the WRF WFP in Section 4.

## 2 Data and Methods

### 2.1 Observations

The 2013 Crop Wind Energy eXperiment (CWEX-13) took place in central Iowa at a 200-turbine wind farm to quantify far-wake impacts of multiple rows of turbines (Lundquist et al., 2014). In CWEX-13, measurements from seven surface flux stations, a radiometer, three profiling lidars and a scanning lidar were collected. This campaign was a component of the larger CWEX project, which explored the interactions of wind turbines with crops, surface fluxes and near-surface flows in different atmospheric stability regimes in flat terrain (Rajewski et al., 2013). Research facilitated by the CWEX projects include: diurnal changes in observed turbine wakes (Rhodes and Lundquist, 2013), turbine interactions with moisture and carbon dioxide fluxes (Rajewski et al., 2014), LES modelling of turbine wakes in changing stability regimes (Mirocha et al., 2015), nocturnal low-level jet (LLJ) occurrences (Vanderwende et al., 2015), diurnal changes of the microclimate near wind turbines (Rajewski et al., 2016), multiple-wake interactions (Bodini et al., 2017), the evolution of turbine wakes during the evening transition (Lee and Lundquist, 2017) and coupled mesoscale-microscale modelling (Muñoz-Esparza et al., 2017).

This wind farm consists of 200 wind turbines, represented by the red dots in Fig. 1. Half of the wind turbines in the wind farm are General Electric (GE) 1.5-MW super-long extended (SLE) model, and the other half are GE 1.5-MW extra-long extended (XLE) model (Rajewski et al., 2013). The cut-in and cut-out speeds of the SLE model are 3.5 and 25 m s$^{-1}$ respectively, and the rated speed is 14 m s$^{-1}$. With the same cut-in speed, the XLE model has lower rated and cut-out wind speeds at 11.5 and 20 m s$^{-1}$. The hub height of both models is 80 m; the rotor diameters of the SLE and the XLE model are 77 and 82.5 m respectively. For simplicity, references to the rotor diameter (D) herein refer to the 77-m rotor diameter.

Power generated by each turbine is recorded by the Supervisory Control and Data Acquisition (SCADA) system every 10 minutes, and we sum up the power production of all turbines for wind-farm production for each 10-min period.

Observations of the wind profile are collected by a profiling lidar and a scanning lidar. The WINDCUBE v1 (WC) profiling lidar (yellow square in Fig. 1) is located 528 m, or 6.3 D, south of the nearest turbine. The WC lidar measures winds at about 0.25 Hz from 40 to 220 m above ground level (AGL) every 20 m via the Doppler beam swinging (DBS) method. The WC lidar derives wind components by measuring radial velocities using DBS at an azimuth angle of 28°. Note that the WC-observed turbulence parameters, namely the turbulence kinetic energy (TKE) and the turbulence intensity (TI), are derived from the variances of the three wind components in 2-min intervals, hence not representing small-scale turbulence. The turbulence parameters are defined by:

$$TKE = \frac{1}{2}(\sigma_u^2 + \sigma_v^2 + \sigma_w^2), \tag{1}$$

$$TI = \frac{\sqrt{\sigma_u^2 + \sigma_v^2}}{\bar{U}}, \tag{2}$$

where $\sigma^2$ are the 2-min averaged variances of the $u$, $v$, and $w$ wind components, and $\bar{U}$ is the mean horizontal WS (Stull, 1988). In CWEX-11, wind-turbine wake measurements at a different location in this wind farm were collected with these instruments (Rhodes and Lundquist 2013), and the errors in the WC lidar measurements due to inhomogeneous flow were explored by Bingöl et al. (2009) and Lundquist et al. (2015).

The WINDCUBE 200S scanning lidar (green square in Fig. 1) is positioned 437 m, or 5.7 D, north of the nearest turbine row. In CWEX-13, the 200S lidar scanning strategy included velocity azimuth display (VAD) scans that measures winds from ~100 to ~4800 m AGL nominally every 50 m for every 3 minutes. In this study, we use the 200S 75-degree-elevation scans (Vanderwende et al., 2015) to estimate horizontal winds every 30 minutes to verify the simulated winds in the boundary layer. In the case study chosen, the dominant WD is south-easterly to south-westerly (Vanderwende et al., 2015), thus some of the 200S measurements below the rotor top (about 120 m AGL) could be influenced by turbine wakes during conditions when the wakes persist longer than 5 D downwind from the turbine (Bodini et al., 2017). On the other hand, the WC measurements are largely unaffected by turbine wakes except when WD is east of 150°. The closest upwind turbine during this simulation period was located over 2.7 km (33 D) to the southeast.

The measurements from the surface flux station can also quantify model skill. The surface flux station of interest (purple square in Fig. 1) is located 681 m, or 8.8 D, south of the closest turbine. At 8 m AGL, the station measures 20-Hz winds via a CSAT3 sonic anemometer, as well as virtual temperature and water-vapour density via a HMP45C probe. After tilt correction (Wilczak et al., 2001), we calculate surface sensible heat flux using a 30-min averaging time period. We use the Obukhov length ($L$) to categorize atmospheric stability conditions:

$$L = -\frac{\overline{T_v} u_*^3}{kg(\overline{w'T_v'})_s}, \tag{3}$$

where $\overline{T_v}$ is the mean virtual temperature, $u_*$ is the frictional velocity, $k$ is the von Karman constant, $g$ is the gravity acceleration, and $(\overline{w'T_v'})_s$ is the surface virtual temperature flux calculated from the 20-Hz measurements (Stull, 1988). A

positive surface sensible heat flux and Obukhov Length ratio ($z L^{-1}$), where $z$ is 8 m, indicates a stable atmosphere, whereas a negative ratio represents unstable conditions.

From 24-27 August 2013, nocturnal LLJs were observed (Vanderwende et al., 2015). No major synoptic events affected the area during this period. Moreover, when the near-surface flows are southerly, the WC and the surface flux station measure winds unaffected by wind turbines (Muñoz-Esparza et al., 2017). Additionally, no curtailment of wind turbines occurred and the instruments operated normally during the period, making these four days ideal for model verification.

## 2.2 Modelling

To establish direct comparison with the observations, we simulate winds with and without the wind farm parameterization (WFP) using the Advanced Research WRF (ARW) model (version 3.8.1) (Skamarock and Klemp, 2008). We simulate the winds on each day separately, from 0 UTC to 0 UTC, after 12 h of spin-up time. The ERA-interim (Dee et al., 2011) and the 0.5° Global Forecast System (GFS) reanalysis datasets provide boundary conditions for two different sets of model runs. We set three domains in our simulations with horizontal resolutions of 9, 3 and 1 km respectively, where the finest domain covers the state of Iowa (Fig. 1). To capture the westerly synoptic flow and the southerly near-surface winds, we position the inner grids northeast of the centres of the coarser grids.

The WFP scheme simulates wind farms and their meteorological influences to the atmosphere. We provide a brief summary here, and the details are discussed in Fitch et al. (2012). Wind turbines slow down ambient wind flow and convert a part of the kinetic energy of wind into electrical energy. The WFP represents this wind-turbine drag force as the kinetic energy harvested by the turbine from the atmosphere:

$$\boldsymbol{F}_{drag} = \frac{1}{2} C_T(|\boldsymbol{V}|)\rho|\boldsymbol{V}|A\boldsymbol{V}, \qquad (4)$$

where $C_T$ is the turbine-specific thrust coefficient (discussed in detail in Fitch, 2015), $\boldsymbol{V}$ is the horizontal velocity vector, $\rho$ is air density, $A = \frac{\pi}{4}D^2$ and is the cross-sectional rotor area, and $D$ is the rotor diameter. This kinetic-energy extraction also causes changes in the atmosphere, namely the kinetic energy loss in the grid cell, which is described by the momentum tendency:

$$\frac{\partial |\boldsymbol{V}|_{ijk}}{\partial t} = \frac{N_t^{ij} C_T(|\boldsymbol{V}|_{ijk})|\boldsymbol{V}|_{ijk}^2 A_{ijk}}{2(z_{k+1}-z_k)}, \qquad (5)$$

where $i$, $j$, and $k$ represents the zonal, meridional, and vertical grid indices, $N_t^{ij}$ is the number of wind turbines per square meter, and $z_k$ is the height at model level $k$. Of the kinetic energy extracted by the turbines, the WFP accounts for the electricity generation with:

$$\frac{\partial P_{ijk}}{\partial t} = \frac{N_t^{ij} C_P(|\boldsymbol{V}|_{ijk})|\boldsymbol{V}|_{ijk}^3 A_{ijk}}{2(z_{k+1}-z_k)}, \qquad (6)$$

where $P_{ijk}$ is the power output in the grid cell in Watts, and $C_P$ is the power coefficient. Assuming negligible mechanical and electrical losses, the rest of the kinetic energy harvested turns into TKE:

$$\frac{\partial TKE_{ijk}}{\partial t} = \frac{N_t^{ij} C_{TKE}(|\boldsymbol{V}|_{ijk})|\boldsymbol{V}|_{ijk}^3 A_{ijk}}{2(z_{k+1}-z_k)}, \tag{7}$$

where $TKE_{ijk}$ is the TKE in the grid cell, and $C_{TKE}$ is the difference between $C_T$ and $C_P$.

In this study, we employ two resolutions of vertical grids: nominally 12-m and 22-m resolution below 400 m AGL, with 80 and 70 total levels respectively (Fig. 2). Three and six vertical levels intersect the atmosphere below and within the rotor layer in the finer vertical grid, while the 22-m grid only allows one full level below and four levels within the rotor layer (Fig. 2). The vertical levels are further stretched beyond the boundary layer. In past research involving the WRF WFP scheme, the selections of vertical resolution within the rotor layer include: 9 to 18 m in Vanderwende et al. (2016); about 10 to 16 m in Volker et al. (2015); about 15 m in Fitch et al. (2012), Fitch et al. (2013a), Fitch et al. (2013b) and Vanderwende and Lundquist (2016); about 20 m in Miller et al. (2015) and Vautard et al. (2014); about 22 m in Lee and Lundquist (2017); about 40 m in Eriksson et al. (2015) and Jiménez et al. (2015).

Moreover, the Mellor-Yamada-Nakanishi-Niino (MYNN) Level 2.5 Planetary Boundary Layer (PBL) scheme is required for the WFP in the WRF model version 3.8.1 (Fitch et al., 2012). Note that substantial upgrades were made on the MYNN PBL schemes in WRF version 3.8 (WRF-ARW, 2016). The MYNN PBL scheme supports TKE advection, active coupling to radiation, cloud mixing from Ito et al. (2015), and mixing of scalar fields. The MYNN scheme also uses the cloud probability density function from Chaboureau and Bechtold (2002), and here we keep the mass-flux scheme deactivated. We summarize the other model configuration details in Table 1.

After verifying the background flow simulated by the WRF model (first 4 rows in Table 2), virtual turbines are added via the WFP (last 4 rows in Table 2). We simulate all the turbines using the 1.5-MW PSU generic turbine model (Schmitz, 2012), in which its specifications are based on the GE 1.5-MW SLE model installed at the wind farm. The turbines within the WRF grid cells are located using the latitudes and longitudes provided by the wind-farm owner-operator. The model grid cells within the wind farm, containing 1 to 4 wind turbines per cell, are labelled as blue numbers in Fig. 1. With the WFP activated, the model simulates the total power production at each time step in each turbine-containing grid cell, regardless of the number of turbines per cell. To match the 10-min average power data from the turbines, we sample 10-min power from the WFP output.

We also estimate the power generation of the WRF simulations without using the WFP. Based on the ambient WS of the turbine-containing grid cells in the control WRF runs, we use the turbine power curve to obtain an assessment of the power every 10 minutes. We then multiply the power with the number of turbines per cell to calculate power in each grid cell, as would be done in wind-energy forecasting without a wake parameterization. This method of power estimation omits wake effects, in contrast to the WFP.

# 3 Results

## 3.1 Ambient Flow Evaluation

The WRF-model simulations without the WFP simulate accurate ambient winds as compared to the lidar measurements. Qualitatively, the ERA12 simulation (see Table 2 for a listing of all the simulations) has skill in simulating WS and WD during the 4-day period, including the occurrence, the strength and the elevation of the nocturnal LLJs (Fig. 3). The 200S records the vertical shear caused by LLJs above 100 m (Fig. 3a), and the WC measures the near-surface winds with high temporal resolution (Fig. 3b). In the observations and the simulations of WS (Fig. 3c), the night-time WS profile is stratified whereas the daytime atmosphere is well-mixed. The WD simulations also match well with the measurements, where in the evening the winds veer, or turn clockwise with height (Fig. 4), while the WD remains relatively constant with height during daytime. Except for the last hours on 24 August, the ERA12 captures the general temporal and vertical fluctuations in WS and WD, when the winds change from south-easterly to south-westerly (Fig. 3 and 4). The 200S measurements above the rotor layer (120 m) are unaffected by turbine wakes (Fig. 3a and Fig. 4a); the LLJs observed above the rotor layer resemble those from the ERA12, confirming the skills of the simulations. To evaluate the effects of boundary conditions and vertical resolutions on simulating winds, we compare the 4 no-WFP runs: ERA12, ERA22, GFS12 and GFS22.

Quantitatively, simulations using finer vertical resolution have more skill in simulating winds than those with coarser resolution (Table 3). In comparison to the 200S and WC observations, the mean absolute errors in WS and WD of the 12-m runs are lower than those of the 22-m runs over the 4-day period, by 0.3 m s$^{-1}$ and 0.8° in average. Particularly in the ERA12, the errors in WS decrease by at least 19% relative to the ERA22. Although the GFS22 yields smaller WS errors than the ERA22, refining the vertical grid of the simulations using either boundary condition dataset improves the WS-prediction skill of the WRF model more than changing the boundary conditions (Table 3). The errors in simulating WD remain similar regardless of the choice of boundary condition or vertical grid. Of all our control runs, the ERA12 simulates the most accurate inflow.

## 3.2 Power Simulations

The simulation omitting the WFP ignores the wake effects on power production of downwind turbines, and therefore overestimates total power. For each 10-min time step, we compare the spatial distribution of power production as well as the total power between the ERA12, the ERA12WF, and the observations; Fig. 5 represents one 10-min time step in the 4-day period. As mentioned above, we calculate the power estimates of ERA12 using the ambient WS, the number of turbines in each grid cell, and the power curve (Fig. 5a). The WRF WFP generates power predictions (Fig. 5b), and we sum up the actual power production in each grid cell (Fig. 5c). We present the total 10-min simulated and observed power of the whole wind farm at the bottom of each panel in Fig. 5, and the total power production of the WFP run matches the observed. We then assemble the 576 10-min total power values over the 4-day period and compare the simulations to the observations (Fig.

6). We also calculate an error and a bias of modelled total power for each 10-min interval, summarizing as the daily root-mean-squared errors (RMSE) and average biases in Table 4 and 5. The large average biases in Table 5 highlight the consistent power overestimation of the no-WFP runs.

220       Over the 4-day period, the WFP produces total power of the whole wind farm that generally agrees with observation (Fig. 6c). Although the RMSEs between the no-WFP and WFP runs are comparable (Table 4), the average biases are smaller in the WFP simulations (Table 5). For instance, the ERA12WF slightly under-predicts total power by -4.9 MW on average (Fig. 6c and Table 5). The ERA12, by contrast, consistently over-predicts power production by 41.5 MW (Fig. 6a and Table 5). The daily positive biases of the ERA12 in the first 2 days are nearly 20% of maximum wind farm production (Table 5).

The average positive power bias of 36.2 MW in the ERA22 is also remarkably larger than the mild negative bias of -15.1 MW in the ERA22WF (Fig. 6b and d, Table 5). Furthermore, the ERA12 and the GFS12 generally outperform the ERA22 and the GFS22 in power predictions, particularly in RMSE (Fig. 6 and Table 5). However, on the last day, with more south-westerly flow, the ERA12 and the ERA22 outperform the ERA12WF and the ERA22WF, while the GFS12WF and the GFS22WF yield smaller errors and biases (Table 4 and 5). Nonetheless, in aggregate, the simulations using the WFP predict

wind-farm power production with more skill than simulations without the WFP.

       As demonstrated by the average absolute errors (Table 3), the WFP power simulations improve when using 12-m rather than 22-m vertical resolution (Fig. 6). Changing the vertical grid improves the predictions more than changing boundary conditions (Table 4 and 5). Particularly in the ERA-interim simulations, the RMSE each day decreases by 19% to 39% when switching from ERA22WF to ERA12WF (Table 4; Fig. 6c and d). Since the power-prediction skills of the ERA-interim-

initiated runs and the GFS-initiated runs are comparable, the rest of the paper will focus on the WFP runs using the ERA-interim as initial and boundary conditions.

       Moreover, to statistically differentiate the power productions from various model runs, we apply the 2-sample Student's t-test. The null hypothesis of a 2-sample t-test is that the two population means are the same, assuming the underlying distributions are Gaussian (Wilks, 2011). Hence, if the resultant p-value is equal to or below 0.05, the two

distributions are statically significantly different at the 95% confidence level. For example, the difference between the 4-day power-production averages from the ERA12 and from the ERA12WF is -46.8 MW, and the respective p-value is 0 (Table 6). Thus the difference between the means is statistically significant. In other words, the ERA12 and the ERA12WF yield different power-production distributions at any confidence level. Similarly, the GFS12 and the GFS12WF lead to statistically different power outputs as the p-value from t-test is 0 as well (Table 7). We also use the 2-sample t-test to contrast the actual

and the modelled power distributions. For instance, all the p-values between the no-WFP runs and the observation are 0, implying those simulations yield power-generation distributions significantly different from the reality (Table 8).

       Given the utility of the WFP, assessing the interactions between atmospheric forcing and power production is an important step to further examine the performance of the WFP. As with the ERA12, the ERA12WF adequately simulates the evolution of the meteorological variables over the 4-day period (Fig. 7a to d). Both the ERA12 and the ERA12WF capture

the overall trends of hub-height ambient WS and WD measured by the WC (Fig. 7a and b), corresponding to Fig. 3 and 4.

On the other hand, although the simulations suggest stronger TKE diurnal cycles than the observations, especially in the first 36 h, the simulated values follow the trends of the WC-measured TKE (Fig. 7c). Although the magnitudes of the surface sensible heat flux of the surface flux station and the simulations differ, their signs change at similar times, particularly in the last three days (Fig. 7d). Hence the WRF model is capable to represent diurnal atmospheric stability changes. Note that in
Fig. 7c, the lidar derives TKE using 2-min variances, which is intrinsically different from the modelled TKE, as discussed in Kumer et al. (2016) and Rhodes and Lundquist (2013). Hence, readers should focus on the general trends of the TKE time series, rather than their absolute values.

The observed WS fluctuates more than the mesoscale-simulated WS during daytime (Fig. 7a). The ramp events, when the WS changes rapidly in a short period (Kamath, 2010; Potter et al., 2009), induce considerable swings in power
production (Fig. 7e). The five distinct ramp events with sudden WS increases are from 00 to 01 UTC on 24 August, from 18 to 19 UTC 24 August, from 00 to 01 UTC 25 August, from 00 to 02 UTC 26 August, and from 00 to 02 UTC 27 August. Most of the ramp events are related to the LLJs (Fig. 3), and the simulated WS usually lags that observed (Fig. 7a). Therefore, the WFP under-predicts total power in nearly all the ramp events (Fig. 7e). Note that the measured WS ranges between the cut-in and rated speed of the wind turbine, when power production is highly sensitive to WS. The strong linkage
between the temporal fluctuations of WS and power emphasizes the importance of accurate WS predictions.

Along the same line, the WFP power performance changes in different meteorological conditions. To quantify WFP's skills, we use the bias in total power as a benchmark, calculated by subtracting the observed power from the WFP simulated power every 10 minutes (Fig. 8). Particularly in conditions of strong winds and weak turbulence, the WFP overestimates wake effects and thus underestimates power. On the other hand, for calm conditions with moderate or strong turbulence, the
WFP tends to underestimate wake effects and thereby over-predicts power (Fig. 8a and c). Besides, the Pearson correlation coefficient between total power bias and WC-observed TKE is 0.48 (not shown).

On the contrary, WD and atmospheric stability have weaker influence on the skill of the WFP in general. The winds gradually rotate from south-easterly to south-westerly over this 4-day period while maintaining similar magnitudes of WS. During this direction shift, the WFP demonstrates a weak positive power bias when the WD is strictly southerly, while the
biases skew negative when the winds have more easterly or westerly component (Fig. 8b). Similarly, the WFP power bias is generally unresponsive to stability changes, although biases tend to be small in strongly stable conditions (Fig. 8d). Moreover, strongly stable conditions tend to have stronger and more distinct wakes (Abkar and Porté-Agel, 2015b; Lee and Lundquist, 2017; Magnusson and Smedman, 1994; Rhodes and Lundquist, 2013).

To isolate the WFP errors in power predictions from the WRF model errors in simulating ambient winds, we analyse a
subset of data where the winds are simulated accurately. When the absolute error in WS is smaller than 1 m s$^{-1}$ and the absolute error in WD is smaller than 5°, the relationships between power bias and WS, WD and TI (Fig. 9a to c) remain similar to the general trends shown in Fig. 8a to c. The WS-power-bias and TI-power-bias correlations become stronger in this subset (Fig 9a and c), compare to the correlations using all the data in the 4-day period (Fig 8a and c). Moreover, when considering only cases of accurate wind predictions, the correlation between power bias and stability increases from -0.06

(Fig. 8d) to -0.42 (Fig. 9d). In the few (27 10-min time steps) unstable conditions with accurate WS predictions, the power bias is generally positive, given moderate WS and high TI (Fig 9 a, c and d). In the stable regime, the WFP tends to underestimate power, regardless of WD (Fig. 9 b and d): 106 of the 125 stable data points are negatively biased. If the few strongest stability points ($z L^{-1}$ larger than 0.55) are removed from the subset shown in Fig. 9d, a weakly negative correlation between power bias and stability emerges as the Pearson correlation coefficient becomes -0.61. Additionally, generally south

to south-westerly flows yield stronger negative power biases (Fig 9).

As expected, when the model properly simulates ambient WS, the WFP performs better. When the ERA12WF predicts larger WS than observed, the simulation over-predicts the total power. The positive WFP power bias corresponds to WS overestimation, and the negative bias is associated with WS underestimation (Fig. 10). Interestingly, when the error in simulated total power lies between $\pm30$ MW, the error of the simulated WS is mostly within $\pm2$ m s$^{-1}$ (Fig. 10). On the other

hand, the power bias does not seem to be related to WD or to ambient TKE: the correlation between the power bias and the simulated WD (TKE) bias is low, at 0.3 (0.22) (not shown). Although the simulated WD and TKE generally match the WC observations (Fig. 7b and c), and the model's skills in simulating WD and TKE are relatively irrelevant to the WFP's power performance.

Although the WFP omits sub-grid-scale wake interactions between the wakes of multiple turbines within a cell, this

omission does not affect the accuracy of the ERA12WF in power prediction: the performance of the WFP is insensitive to the number of turbines per model grid cell. The turbine-normalized bias demonstrates no dependence on the number of turbines within the model grid cell (Fig. 11). Each whisker in Fig. 11 marks the maximum, the upper quartile, the median, the lower quartile, and the minimum of the average bias. Despite the large positive biases of the maxima, more than half of the average biases fall between $\pm1.5$ MW, regardless of the numbers of turbines per cell (Fig. 11). Simulating 1 or 4 turbines

in a grid cell (Fig. 1) does not influence the WFP's overall power-prediction performance in the cases shown here.

Furthermore, the WFP performance remains consistent between upwind and downwind turbines, based on their positions against the ambient winds (Fig. 12). Given the square shape of grid cells, we determine the sequential rows of turbines during strictly southerly flows, with WD between 175° and 185° (Fig. 12a). The bulk of the normalized power biases fall within 0 to 0.4 MW, regardless of the upwind-downwind positions of turbines (Fig. 12b). Additionally, the power

bias is independent of the mean distance between the actual turbine locations and the centre points of their respective grid cells (not shown).

**4 Discussion**

Herein, we compare WRF model simulations with different choices of vertical resolutions and boundary conditions. The evidence suggests that, at least for this onshore case with a strong diurnal cycle, the vertical resolution is more crucial

than the choice of boundary conditions in simulating accurate winds and wind-power production. Shin et al. (2011) have explored the impacts of the lowest model level on the performance of various PBL schemes in the WRF model, suggesting

that increasing the number of model layers can simulate more accurate surface layer in different stability regimes. In this study, we further illustrate that establishing more vertical levels in the boundary layer as well as the rotor layer improves the skills of the WRF model in simulating ambient WS, ambient WD and wind power (Table 3, 4 and 5). Furthermore, Carvalho et al. (2014) discussed the effects of different reanalysis datasets on wind-energy production estimates, and found the ERA-interim presents the most precise initial and boundary conditions, followed by the GFS. Herein, we test the ERA-interim and the 0.5° GFS, and both datasets produce simulations that resemble observed winds and power generations. Since the simulated power is sensitive to the resolution of model vertical grid, particularly near the surface, future WRF WFP users should select vertical levels with care.

Additionally, the outcomes from the statistical tests among the model runs further validate the importance of using the WFP as well as using a fine vertical grid. From the Student's t-test, the p-values of all the no-WFP and WFP pairs are 0 (Table 6 and 7), demonstrating that the differences between the power-generation distributions of the no-WFP runs and the WFP runs are statistically significant at any confidence level. Therefore, to accurately simulate power production, applying the WFP is better than not using it, regardless of the choice of vertical resolution and boundary condition, and the corresponding improvements in Table 4 and 5 are statistically significant. Although the distinction between the GFS12WF and GFS22WF is not statistically significant at the 90% confidence level (Table 7), switching from ERA22WF to ERA12WF improves power simulations significantly at 99% confidence (Table 6). In particular, the RMSE drops by 19.1 MW and the bias reduces by 10.2 MW in average in the ERA12WF (Table 4 and 5), and these are proven statistically significant.

Similarly, results from the statistical tests between the distributions of power from simulations and observations support the value of the WFP applied in a fine vertical grid. The p-values of the ERA12WF-observed pair and the GFS12WF-observed pair are 0.106 and 0.167 respectively (Table 8). The high p-values illustrate the distinctions between the distribution of observed power and the distributions of simulated power from the 12-m WFP simulations are not statistically significant, at the 90% confidence level. Among all the simulations analysed above, running the WFP over the 12-m vertical grid is the only combination that is not statistically different from observations (Table 8). In other words, the 12-m WFP simulations provide the closest approximations to the actual power production, regardless of the boundary-condition dataset.

One of the objectives of this study is to propose general directions for improvements on the WFP. First of all, as the key determinant of wind-power production, WS plays a critical role. Ramp events pose a challenge to the WRF model in simulating WS as well as to the WFP in predicting power (Fig. 7a and e). On the other hand, windy conditions of WS exceeding 10 m s$^{-1}$, although below rated speed, lead to WFP power underestimation (Fig. 8a). Furthermore, the WFP performance depends more on the horizontal winds and turbulence, rather than their vertical components, since the power bias correlates stronger with TI than TKE (Fig. 8c). Reducing turbulence diffusion in the WRF model could potentially yield more accurately simulated winds in stable conditions, including LLJs (Sandu et al., 2013); active research in modifying mixing lengths (Jahn et al., 2017) also suggests promising model improvements. More importantly, sharpening the skills of the WRF model in simulating WS can improve the WFP power performance (Fig. 10). Future versions of the WRF model

and the WFP should aim to better account for instantaneous horizontal WS variations and the subsequent sub-gird wake interactions.

Besides necessary improvements in simulating ambient WS, the WFP scheme itself also requires refinements. When background winds are accurately predicted, the power-bias dependence on WS and TI remain strong (Fig. 9a and c). Moreover, the correlation between the WFP performance and atmospheric stability becomes weakly negative without the

strongly stable data (Fig. 9d). Therefore, even when the simulated winds are close to observations, the WFP tends to underestimate power during high WS, low TI and stable conditions. In contrast, the WFP tends to over-predict power in calm, unstable, and turbulent conditions, with the caveat that a small number of unstable cases are considered here. The WFP scheme appears to overestimate wake loss within a grid cell in stable and windy conditions, and underestimate wake effects in an unstable and well-mixed atmosphere. Certainly the interactions between WD and wind-farm layout affect the power-

bias relationships, and further sensitivity tests can provide more insight into the WFP performance, particularly in intra-cell WS reduction. We demonstrate that inter-cell wake effects are not the critical factor to power error (Fig. 12b), hence the inability of the WFP to simulate intra-cell wake effects can explain the biases when many of the turbines experience accurately-simulated ambient flow.

In contrast, WD has no clear influence on the WFP skill (Fig. 8b) in this case, although the irregular shape of the wind

farm adds uncertainty to this relationship. Similarly, the skill of the WFP for this case is insensitive to the number of virtual turbines per cell, and the downwind position of turbines against inflow (Fig. 11 and 12). Compared to the power overestimation of downwind turbines in the idealized cases described in Vanderwende et al. (2016), both the upwind and downwind turbine-containing cells presented in this study have consistent positive biases on power production (Fig. 12). Our findings suggest that the WFP is skilful in simulating power of aggregate wind turbines and can represent the impact of inter-

cell wakes on power. In the end, the primary limitation of the WFP is rooted in the ambient simulated WS in the WRF model.

**5 Conclusion**

The WFP scheme in the WRF model (version 3.8.1) provides a convenient way to represent wind farms and their meteorological impacts in the NWP models. However, its power predictions have not been verified for onshore wind farms

or in a range of WS conditions. Herein, we evaluate the performance of the WFP in various atmospheric conditions to guide users of the WFP and to suggest future WFP advancements.

Using data from the CWEX-13 campaign, we select a 4-day period, from 24 to 27 August 2013, for our case study, due to the consistent nocturnal LLJ occurrences. We use measurements from a profiling lidar, a scanning lidar and a surface flux station to verify the ambient flows simulated by the WRF model. The wind farm of interest, located in central Iowa, consists

of 200 1.5 MW wind turbines.

We explore the role of vertical resolution in the operation of the WRF WFP. We evaluate two vertical grids with 12-m and 22-m resolution near the surface. We find that the finer vertical resolution produces simulations agree better with observed WS, WD and power than the simulations with coarser vertical resolution. Further, because the WFP accounts for wake effects on power production of downwind turbines, the use of the WFP enables more accurate power prediction, whereas simulations without the WFP generally over-predict power production. Statically, the WFP simulations with a fine vertical grid, regardless of the boundary conditions, are the most skilful in simulating power.

The skill of the WFP varies with meteorological conditions. When the model simulates WS close to the observations, the WFP predicts power properly, making WS the critical factor in improving the WFP. Rapid temporal fluctuations in WS introduce errors in power simulations, especially during ramp events. Further, in windy, stable and less turbulent conditions, the WFP tends to overestimate wake effects and thus underestimates power production. On the other hand, the WFP performance demonstrates no clear dependence on the number of turbines per model grid cell or the downwind distance of turbines with respect to the upwind ones.

In conclusion, we demonstrate the value of the WRF WFP and the importance of using a fine vertical grid. Since WS greatly affects the skill of the WFP, subsequent research could include evaluating the WFP for an even larger range of WS, especially at WS beyond the turbine cut-out speed (which would be 25 m s$^{-1}$ in this case; no such high WS were observed during the CWEX-13 campaign). Evaluating the performance of other wind-farm layouts in locations with complex terrain is also needed. Modifications in the inflow WS considered by the WFP, for example, considering the rotor equivalent wind speed (REWS) (Wagner et al., 2009), may bring promising improvements. More accurate power forecasts will help shaping a more competitive wind-energy industry, and further facilitate grid integration of wind energy (MacDonald et al., 2016).

**Data Availability**

The code of the WRF-ARW model (doi:10.5065/D6MK6B4K) is publicly available at http://www2.mmm.ucar.edu/wrf/ users/download/get_source.html. This work uses the WRF-ARW model and the WRF Preprocessing System (WPS) version 3.8.1 (released on 12 August, 2016), and the wind farm parameterization is distributed therein. The PSU generic 1.5 MW turbine (Schmitz, 2012) is available at doi:10.13140/RG.2.2.22492.18567. The user input required to run the WRF WFP is available at doi:10.5281/zenodo.847780.

**Acknowledgements**

This study was funded by the National Science Foundation (Grant number: 1413980; Project Title: CNH-Ex: Legal, Economic, and Natural Science Analyses of Wind Plant Impacts and Interactions). The CWEX project was supported by the National Science Foundation under the State of Iowa EPSCoR Grant 1101284. The role of the University of Colorado Boulder in CWEX-13 was supported by the National Renewable Energy Laboratory. The authors thank the reviewers and

editors for their thoughtful comments and suggestions. The authors would like to acknowledge the high-performance computing support from Yellowstone (ark:/85065/d7wd3xhc), provided by NCAR's Computational and Information Systems Laboratory, and sponsored by the National Science Foundation. The authors also thank NextEra Energy for providing the wind turbine power data, Iowa State University for providing the surface flux measurements, and NRG Renewable Energy

Systems and Leosphere for providing the 200S scanning lidar used in the CWEX-13 campaign.

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

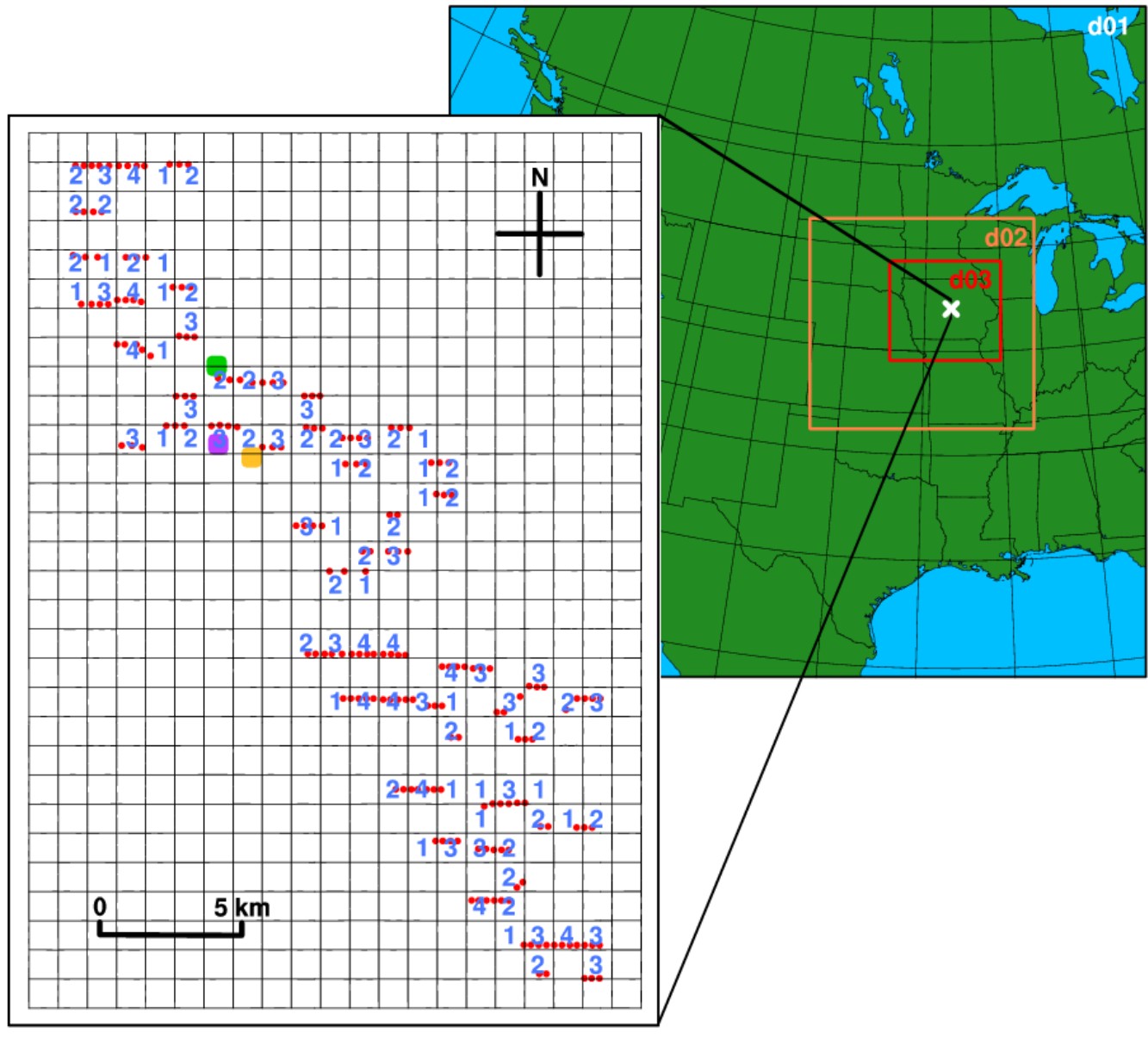

**Figure 1: Map of the 3 domains (d01, d02 and d03) in the WRF simulations (right), with the white x representing the CWEX-13 wind farm. Zoom-in map of the wind farm (left), with the black horizontal and vertical lines outlining the WRF grid cells, the red dots as the actual locations of wind turbines, the blue numbers as the number of wind turbines per WRF grid cell, the yellow square as the WC lidar, the green square as the 200S lidar and the purple square as the surface flux station. Other instruments were deployed in CWEX-13, and only the instruments used herein are shown.**


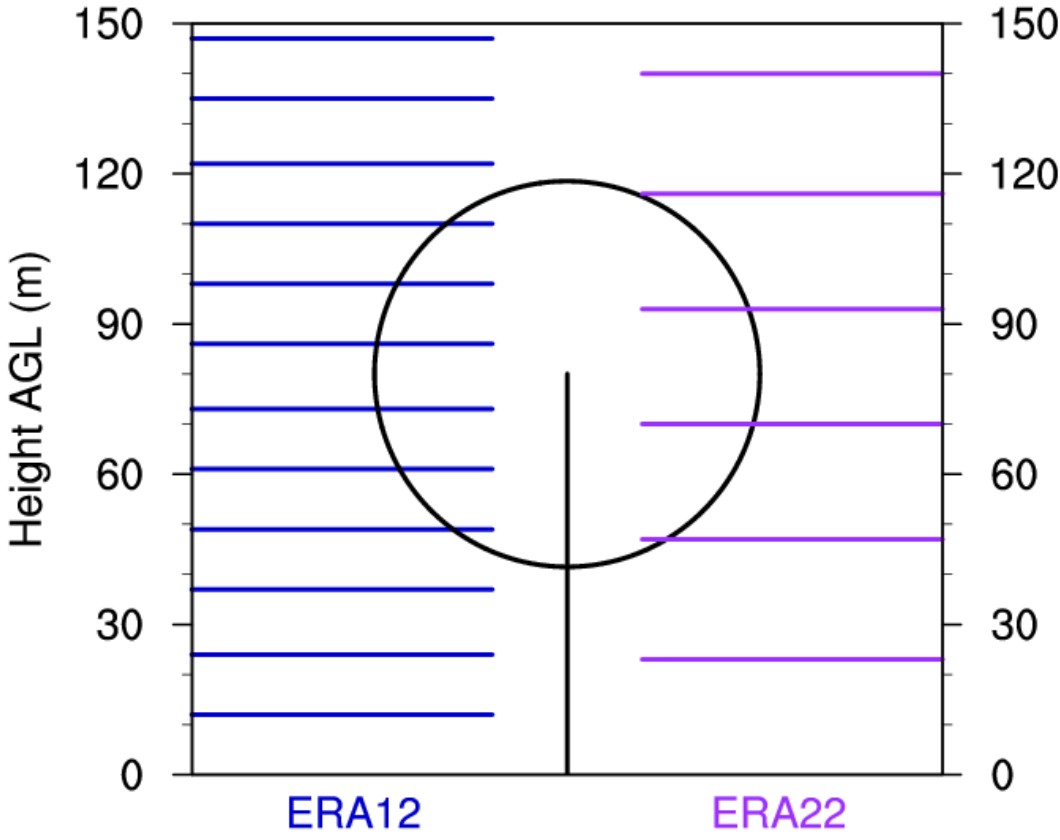

**Figure 2: Illustration of the two vertical grids chosen: the 12-m grid on the left in blue and the 22-m grid on the right in purple.** Both grids shown use the ERA-interim as the boundary conditions. The simulations initiated with the 0.5° GFS have similar vertical grids.

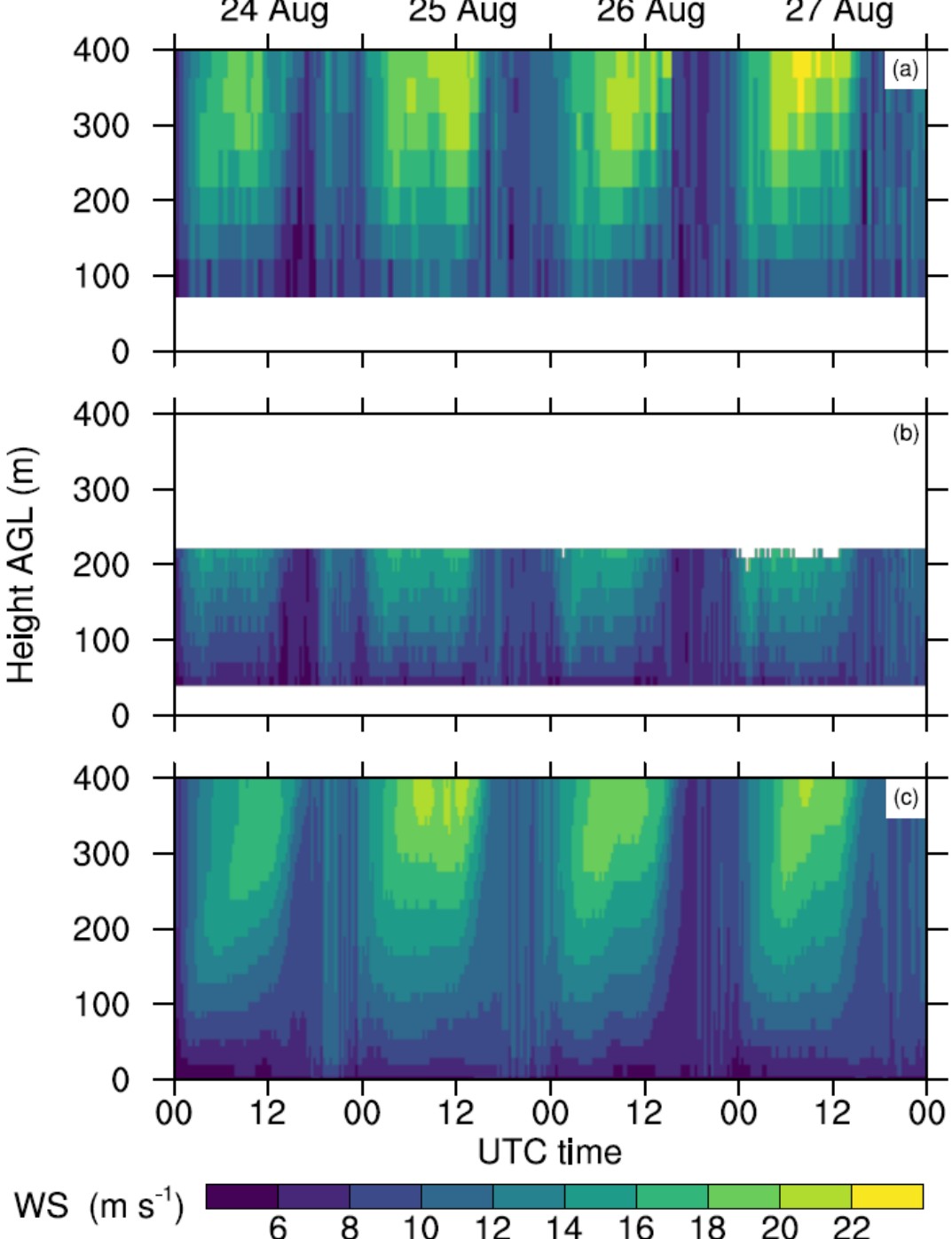

**Figure 3: Time-height contour of WS from the 200S (a), the WC (b) and the ERA12 at the closest grid point to the 200S (c).**


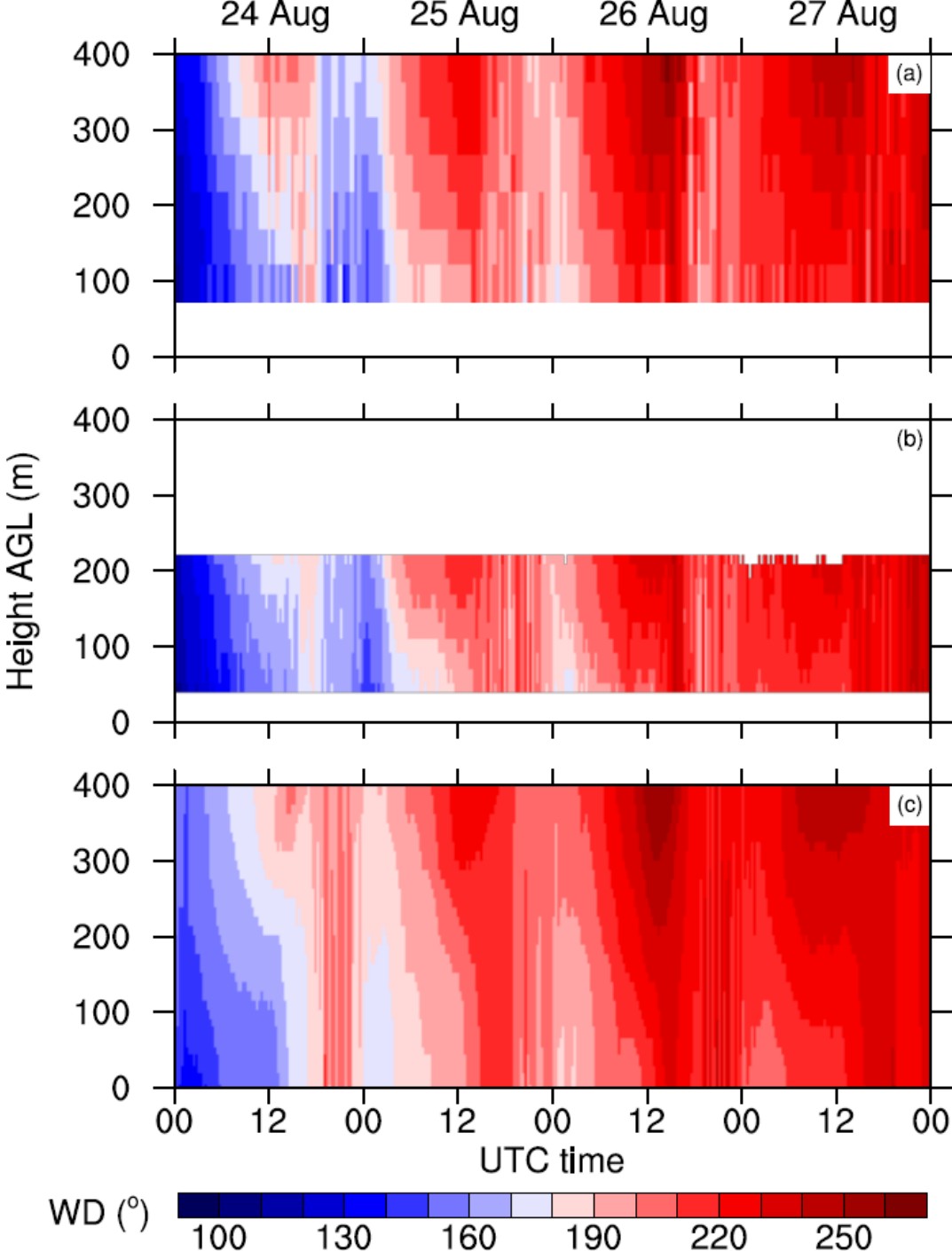

**Figure 4: As in Fig. 3, but for WD.**

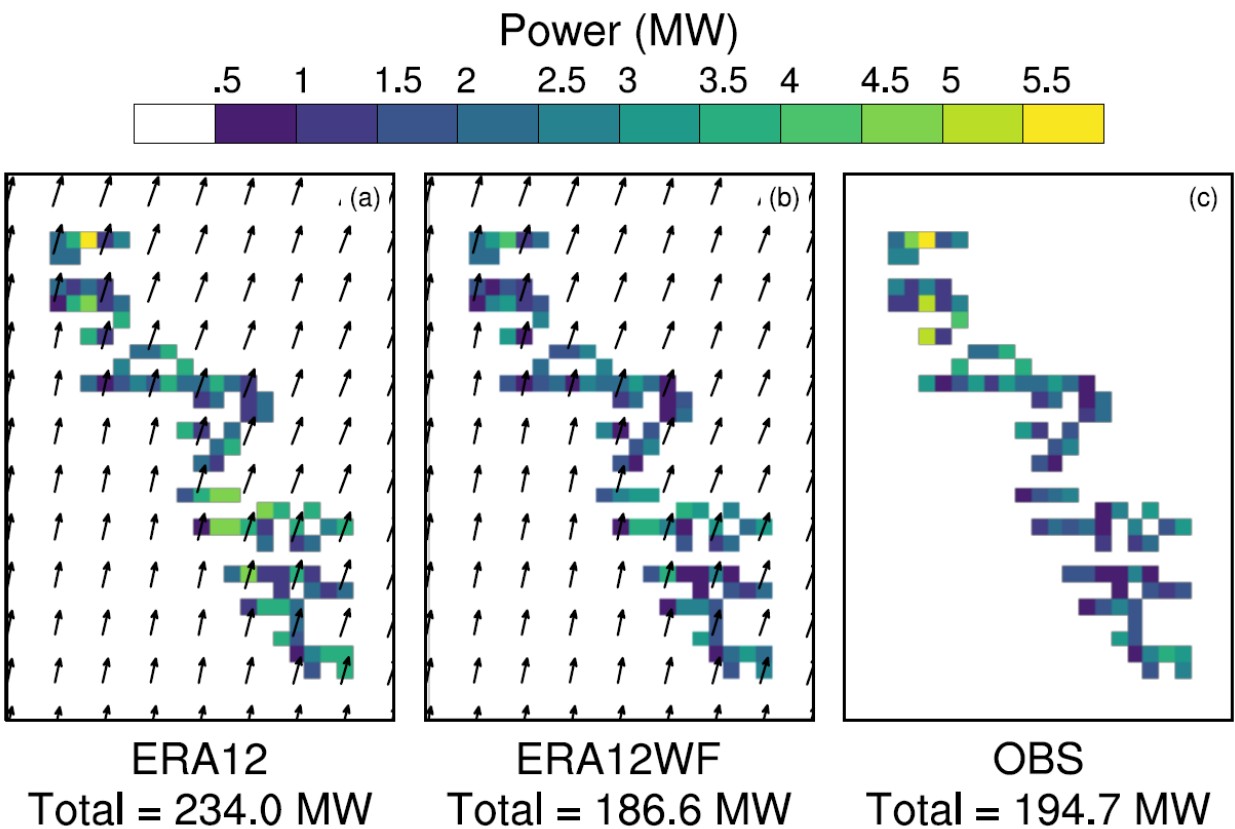

**Figure 5:** The power production for one 10-min period from the ERA12 estimates (a), the ERA12WF outputs (b) and the observation (abbreviated as OBS) (c). The total power in each grid cell is presented regardless of the number of turbines in each cell, and the wind-farm totals are summarized at the bottom. The vectors indicate the simulated winds, and their lengths correspond to the horizontal velocity magnitude.

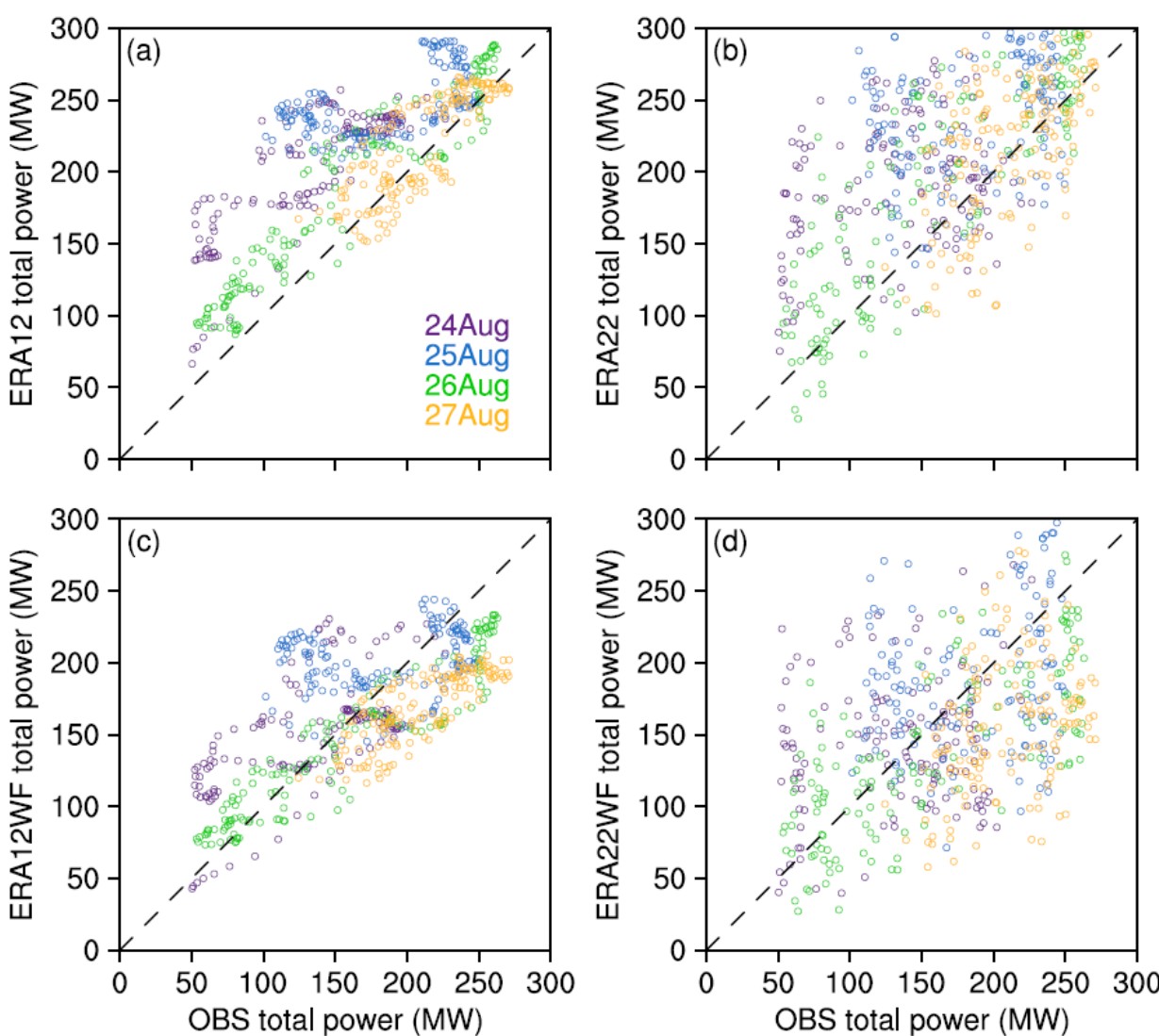

Figure 6: Scatter plots comparing the 10-min average observed total wind-farm power over the 4-day period against the calculated total power from the ERA12 (a) and the ERA22 (b), and the simulated total power from the ERA12WF (c) and the ERA22WF (d). The dots represent the total power productions on 24 August (purple), 25 August (blue), 26 August (green) and 27 August (yellow).

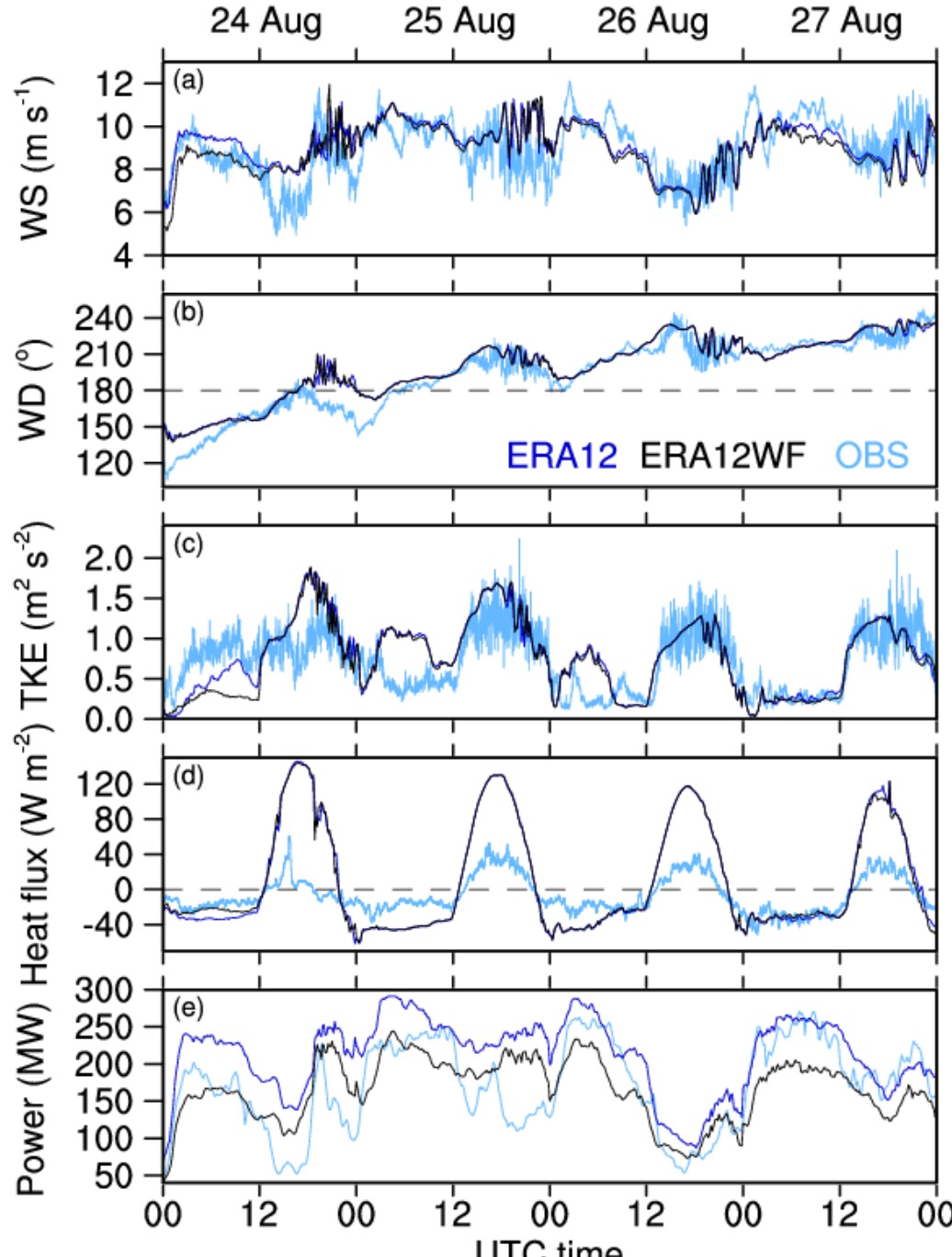

**Figure 7: Time series of hub-height WS (a), hub-height WD (b), hub-height TKE (c), surface sensible heat flux (d), and total wind-farm power (e) from the simulations (ERA12, in blue; ERA12WF, in black) and the measurements (in light blue). The simulated values are interpolated to hub height in the grid cell closest to the WC. In (b), the grey horizontal dash line marks the WD of 180°. In (d), the grey horizontal dash line marks the heat flux of 0 W m⁻².**


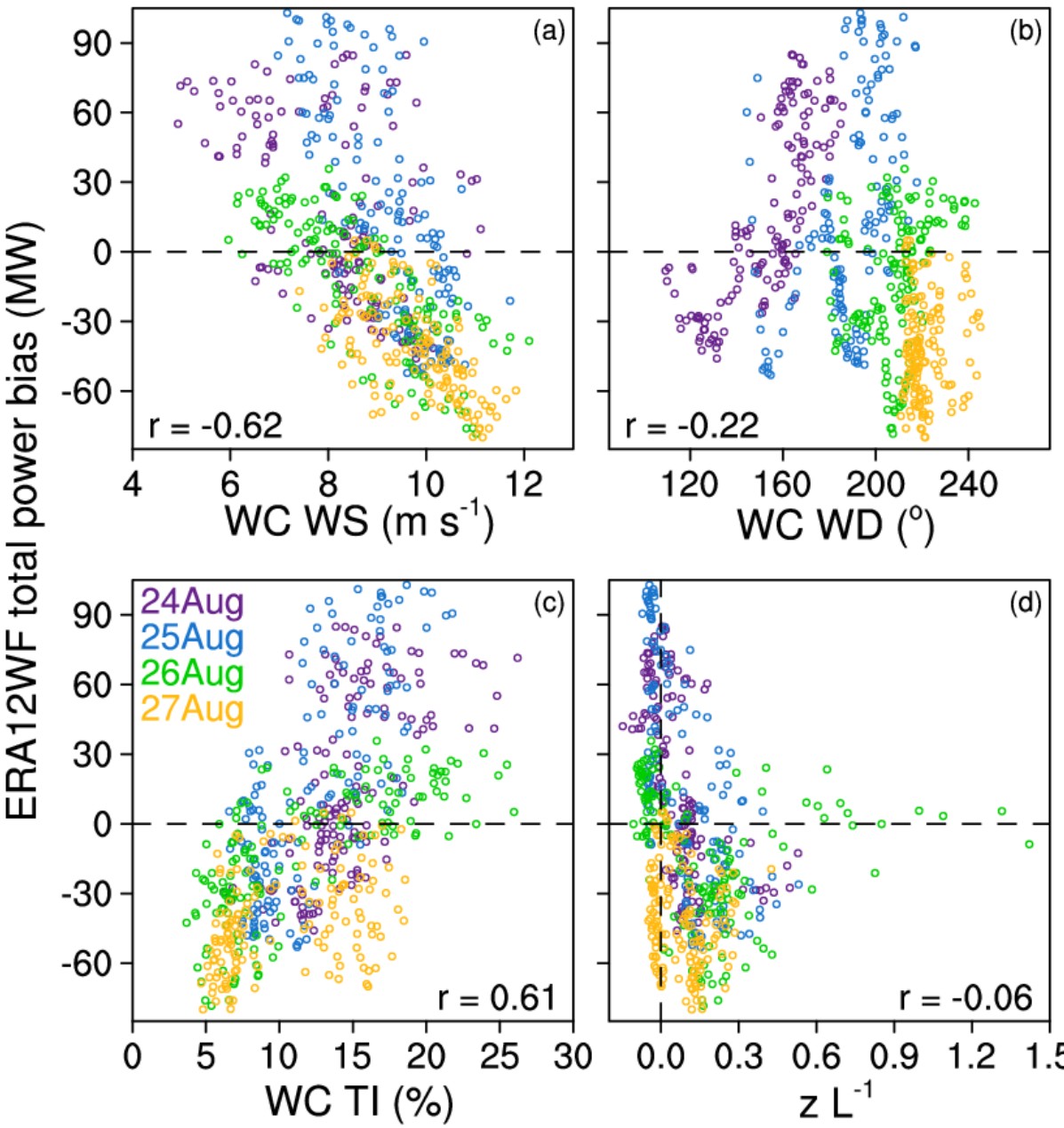


**Figure 8: Scatter plots of the bias of the ERA12WF 10-min total power and the WC-observed hub-height WS (a), hub-height WD (b), hub-height TI (c) and stability parameter $z\,L^{-1}$ measured at the surface flux station (d). The r represents the Pearson correlation coefficient. Similar to Fig. 6, different coloured dots represent biases on different days. The horizontal black dash lines mark the zero power bias. In (d), the vertical black dash line at zero $z\,L^{-1}$ differentiates the two stability regimes.**

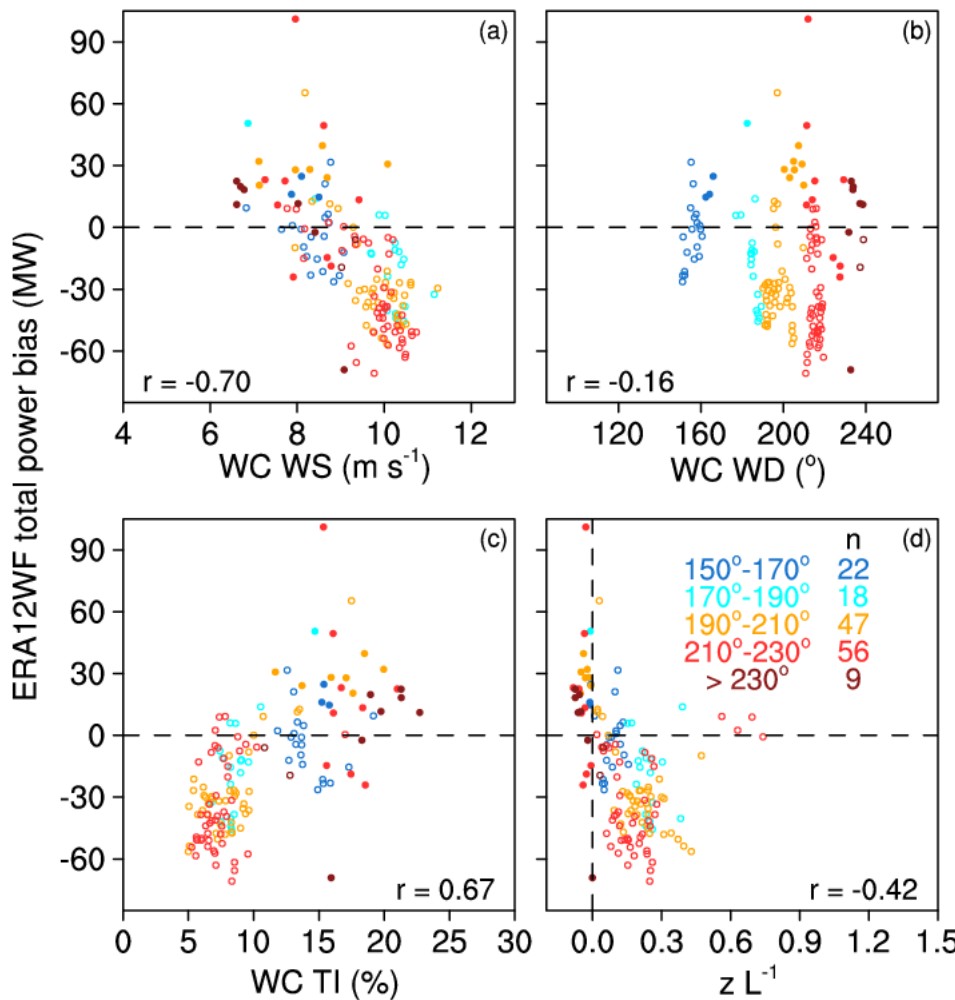


**Figure 9: As in Fig. 8, and only including data when the winds are accurately simulated in the ERA12WF run: the modelled-observed absolute error in WS smaller than 1 m s$^{-1}$ and the absolute error in WD smaller than 5°. Different colours represent different WD bins: 150° to 170° in blue, 170° to 190° in cyan, 190° to 210° in orange, 210° to 230° in red, and 230° and beyond in maroon. The n values illustrate the sample size in each WD bin. Solid circles represent unstable conditions (z L$^{-1}$ smaller than 0)**
**and hollow circles represent stable conditions (z L$^{-1}$ larger than 0).**

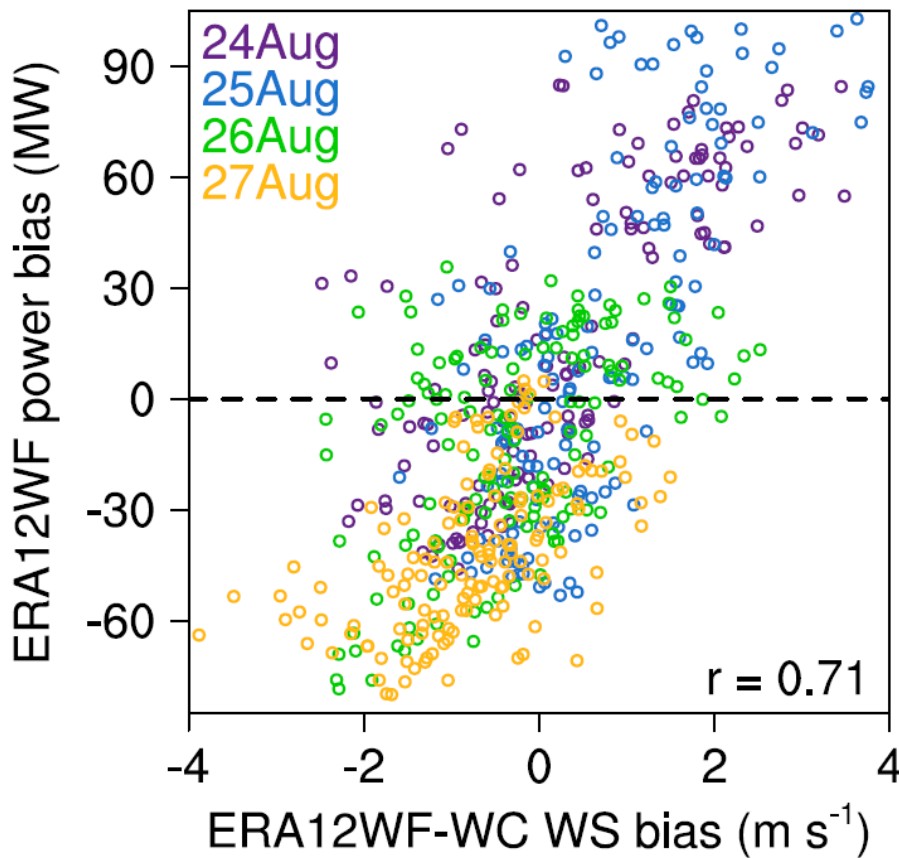

**Figure 10:** Scatter plot between the bias of the ERA12WF 10-min total power compared to observation, and its bias of the simulated hub-height WS in the closest grid cell to the WC. The r represents the Pearson correlation coefficient.


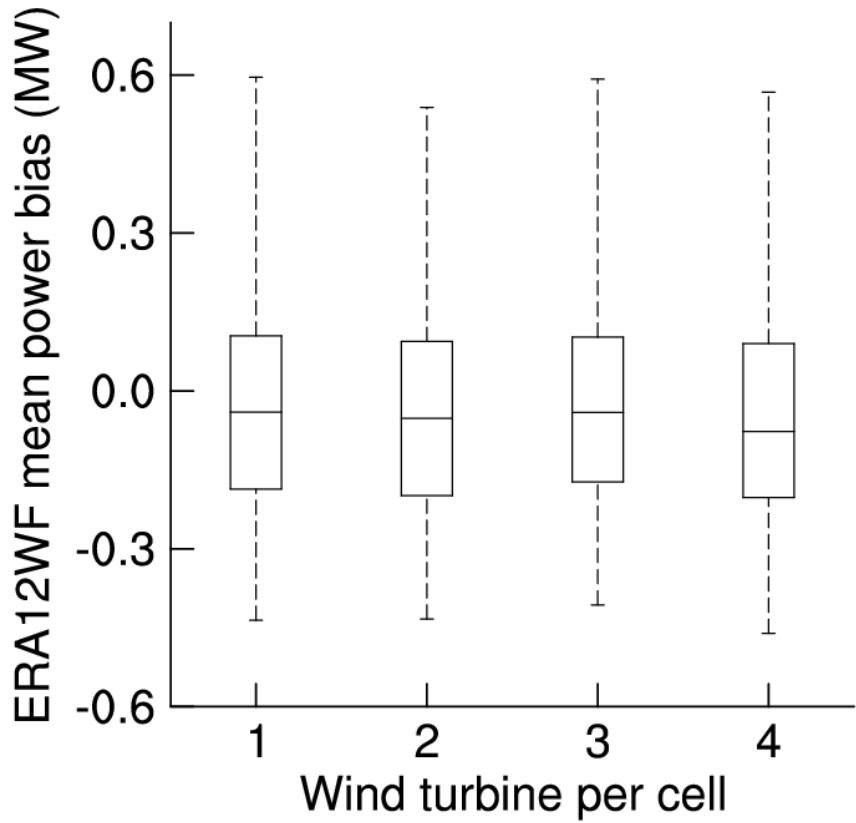

**Figure 11: Boxplot of the average bias of the ERA12WF simulated power across different numbers of wind turbine per WRF grid cell (Fig. 1) every 10 minutes during the 4-day period.**

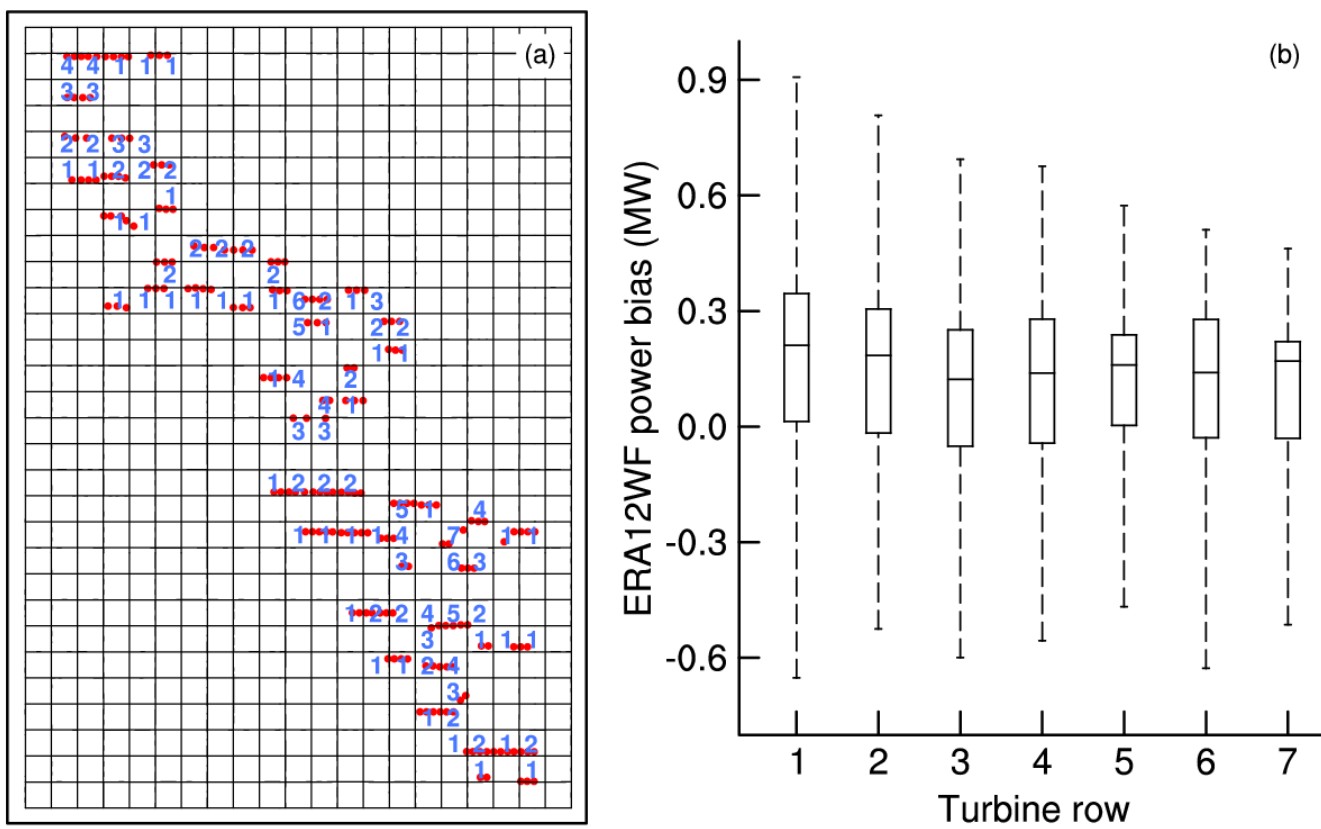


**Figure 12: Map of the wind farm where the blue numbers represent the row number from the upwind row during southerly winds (a). The upwind row number is reset to 1 when the next two downwind grid boxes to the North contain no turbines. Boxplot of the average ERA12WF power bias normalized over different number of wind turbine rows, when the hub-height WD in the grid cell closest to the WC is between 175° and 185° (b).**


**Table 1: The WRF model configuration.**

| Parameterization | Scheme | Reference |
|---|---|---|
| Cumulus | Kain-Fritsch | Kain (2004) |
| Land surface | NOAH LSM | Ek et al. (2003) |
| Land surface roughness | Thermal roughness length | Chen and Zhang (2009) |
| Microphysics | Thompson aerosol-aware | Thompson and Eidhammer (2014) |
| PBL | MYNN Level 2.5 | Nakanishi and Niino (2006) |
| Radiation | RRTMG | Iacono et al. (2008) |

**Table 2: List of WRF simulations and their features.**

| Run name | Boundary condition | Vertical resolution | WFP |
|---|---|---|---|
| ERA12 | ERA-interim | 12 m | No |
| ERA22 | ERA-interim | 22 m | No |
| GFS12 | 0.5° GFS | 12 m | No |
| GFS22 | 0.5° GFS | 22 m | No |
| ERA12WF | ERA-interim | 12 m | Yes |
| ERA22WF | ERA-interim | 22 m | Yes |
| GFS12WF | 0.5° GFS | 12 m | Yes |
| GFS22WF | 0.5° GFS | 22 m | Yes |


**Table 3: Average absolute error in WS (m s$^{-1}$) and WD (°) of different no-WFP runs.**

|  | ERA12 | ERA22 | GFS12 | GFS22 |
|---|---|---|---|---|
| 200S 120 m WS | 1.49 | 1.84 | **1.35** | 1.54 |
| WC 120 m WS | **1.21** | 1.63 | 1.34 | 1.48 |
| WC 80 m WS | **1.24** | 1.64 | 1.36 | 1.55 |
| WC 40 m WS | **1.47** | 1.90 | 1.53 | 1.86 |
| 200S 120 m WD | 14.99 | 15.98 | **14.68** | 14.99 |
| WC 120 m WD | **12.66** | 13.86 | 13.07 | 13.47 |
| WC 80 m WD | **13.23** | 14.55 | 13.85 | 14.24 |
| WC 40 m WD | **14.19** | 15.58 | 14.83 | 15.15 |

The smallest errors across different WRF settings are highlighted in bold.


**Table 4: RMSE of 10-min total power (MW) of different model runs each day.**

|          | 24 Aug | 25 Aug | 26 Aug | 27 Aug | 4-day mean |
|----------|--------|--------|--------|--------|------------|
| ERA12    | 73.6   | 73.5   | **35.4** | **22.6** | **51.3** |
| ERA22    | 79.5   | **72.8** | 48.5 | 41.0   | 60.5       |
| GFS12    | **62.0** | 76.5 | 58.3   | 40.9   | 59.4       |
| GFS22    | 73.9   | 89.6   | 65.3   | 51.9   | 70.2       |
| ERA12WF  | 42.2   | **49.4** | **31.1** | 46.5 | **42.3** |
| ERA22WF  | 61.7   | 61.2   | 50.9   | 71.6   | 61.4       |
| GFS12WF  | 46.2   | 54.6   | 34.1   | **36.1** | 42.8     |
| GFS22WF  | **40.0** | 60.0 | 32.6   | 37.3   | 42.5       |

**Table 5: Average bias of 10-min total power (MW) of different model runs each day.**

|          | 24 Aug | 25 Aug | 26 Aug | 27 Aug | 4-day mean |
|----------|--------|--------|--------|--------|------------|
| ERA12    | 68.3   | 62.6   | **26.8** | 8.1  | 41.5       |
| ERA22    | 58.3   | **52.1** | 28.0 | **6.2** | **36.2** |
| GFS12    | **49.4** | 65.0 | 51.8   | 29.0   | 48.8       |
| GFS22    | 65.5   | 80.7   | 60.3   | 35.8   | 60.6       |
| ERA12WF  | 17.5   | 16.6   | -12.2  | -41.6  | -4.9       |
| ERA22WF  | 10.4   | **0.6** | -17.6 | -53.6  | -15.1      |
| GFS12WF  | 3.8    | 22.2   | **9.6** | -18.6 | **4.3**  |
| GFS22WF  | **2.9** | 29.7 | 10.9   | **-12.3** | 7.8     |


The RMSEs and biases closest to zero across different days are highlighted in bold.

**Table 6: Differences (first value) and p-values (second value) from 2-sample t-tests of simulated power from different ERA runs.**

|  |  | ERA12 | ERA12WF | ERA22WF |
|---|---|---|---|---|
|  | 4-day mean | 41.8 | -4.9 | -15.1 |
| ERA12 | 41.8 |  | -46.7; 0 |  |
| ERA22 | 36.1 | 5.7; 0.03 |  | -51.2; 0 |
| ERA12WF | -4.9 |  |  |  |
| ERA22WF | -15.1 |  | 10.2; $9.6\times10^{-4}$ |  |


**Table 7: As in Table 6, but for GFS runs.**

|  |  | GFS12 | GFS12WF | GFS22WF |
|---|---|---|---|---|
|  | 4-day mean | 48.6 | 4.2 | 7.8 |
| GFS12 | 48.6 |  | -44.4; 0 |  |
| GFS22 | 60.5 | -11.9; $1.1\times10^{-7}$ |  | -52.7; 0 |
| GFS12WF | 4.2 |  |  |  |
| GFS22WF | 7.8 |  | -3.6; 0.16 |  |

**Table 8: P-values from 2-sample t-tests of the 10-min observed power and the 10-min simulated power from different model runs.**

|  | Simulated 4-day mean | Observed 4-day mean | Difference of means | P-value |
|---|---|---|---|---|
| ERA12 | 212.7 |  | 41.8 | 0 |
| ERA22 | 207.0 |  | 36.1 | 0 |
| GFS12 | 219.5 |  | 48.6 | 0 |
| GFS22 | 231.4 | 170.9 | 60.5 | 0 |
| ERA12WF | 166.0 |  | -4.9 | 0.106 |
| ERA22WF | 155.8 |  | -15.1 | $6.5\times10^{-6}$ |
| GFS12WF | 175.1 |  | 4.2 | 0.167 |
| GFS22WF | 178.7 |  | 7.8 | 0.014 |