# Peer review of "Evaluation of the wind farm parameterization in the Weather Research and Forecasting model (version 3.8.1) with meteorological and turbine power data"

_Geoscientific Model Development, 2017_

## Short Comment (SC1) · 8 Jun 2017

Dear authors,

In my role as Executive editor of GMD, I would like to bring to your attention our Editorial version 1.1:

http://www.geosci-model-dev.net/8/3487/2015/gmd-8-3487-2015.html

This highlights some requirements of papers published in GMD, which is also available

on the GMD website in the 'Manuscript Types' section:

http://www.geoscientific-model-development.net/submission/manuscript_types.html

In particular, please note that for your paper, the following requirements have not been met in the Discussions paper:

- "All papers must include a section, at the end of the paper, entitled 'Code availability'. Here, either instructions for obtaining the code, or the reasons why the code is not available should be clearly stated. It is preferred for the code to be uploaded as a supplement or to be made available at a data repository with an associated DOI (digital object identifier) for the exact model version described in the paper. Alternatively, for established models, there may be an existing means of accessing the code through a particular system. In this case, there must exist a means of permanently accessing the precise model version described in the paper. In some cases, authors may prefer to put models on their own website, or to act as a point of contact for obtaining the code. Given the impermanence of websites and email addresses, this is not encouraged, and authors should consider improving the availability with a more permanent arrangement. After the paper is accepted the model archive should be updated to include a link to the GMD paper."

- Inclusion of Code and/or data availability sections is mandatory for all papers and should be located at the end of the article, after the conclusions, and before any appendices or acknowledgments. For more details refer to the code and data policy.

Please add a code availability section stating how to access WRF model version 3.8.1.

Yours,

Astrid Kerkweg

---

## Author Comment (AC1) · 8 Jun 2017

Dear Editor,

Thank you for your comments. Our apologies on not explicitly providing instructions on how to access WRF 3.8.1 in the code availability section. The below will be included in the next revision of the manuscript.

"The code of the WRF-ARW model (doi:10.5065/D6MK6B4K) is publicly available at http://www2.mmm.ucar.edu/wrf/users/download/get_source.html.  This work uses the

WRF-ARW model and the WRF Pre-Processing System (WPS) version 3.8.1 (released on 12 August, 2016), and the wind farm parameterization is distributed therein."

Please kindly let us know if you have any other concerns.

Best regards, Joseph Lee

---

## Referee Comment (RC1) · Anonymous Referee #1 · 23 Jun 2017

This manuscript tries to evaluate the wind farm parametrization in WRF by comparing WRF power results to SCADA power data from a given onshore wind farm in Iowa and from additional lidar observations from the CWEX-13 experiment. Main result of the study is said to be the fact that the simulated ambient wind speed is the most important parameter for the quality of the simulation of the wind farm yield. The subject is important and is expected to have both scientific and economic impact.

Thus, the manuscript deserves publication. But unfortunately – at least to my estimation – there are several issues which should be addressed before a publication could

be recommended.

Major review points

(1) The study presented in this manuscript does not differentiate between deviations due to the simulation of the magnitude of the ambient wind speed and those due to inabilities of the wind farm parametrization (WFP). The WFP is not introduced in any detail and none of the found deviations is attributed to any feature of the WFP.

(2) Mesoscale models such as WRF are known to underrepresent the nocturnal low-level jet phenomenon. This has been analysed and explained by Sandu et al. (2013). Deviations in simulated power due to deviations in simulating the ramp effects at the onset of low-level jets (LLJ) have to be attributed to WRF itself and not to WFP. With regard to this known feature it seems a bit unlucky to choose a LLJ episode for this WFP evaluation.

(3) The lower right frame of Fig. 8 shows the dependence of the bias of the simulated power output from atmospheric stability. The authors interpret this figure as showing no significant dependence. My impression is, if the very few data points beyond the stability of 0.6 are skipped, that there is a significant influence of atmospheric stability (leading to a negative bias for more stable situations).

(4) The discussion section (Section 4) makes reference to several results which have not been shown in the preceding results section (Section 3). Therefore, the reader cannot prove these conclusions.

(5) The main conclusion that simulations with WFP are better than those without it is quite trivial.

(6) The study does not present any points which would allow for an enhancement of the simulation tool (either WRF or WFP or both). (Please refer to comment (1) above as well)

(7) sigma_u and sigma_w cannot be derived from a single monostatic remote sensing

device (be it a sodar or a lidar) operating in Doppler beam swinging mode. Thus, the variables TKE and TI (equations (1) and (2)) are very unsure and cannot be used for a reliable evaluation study. This problem can be easily seen from the massive scatter of the TKE values in the third frame of Fig. 7.

Minor review points

(8) The analytic wind farm model in Emeis (2010) is not based on an exaggerated surface roughness. This model uses the farm-averaged thrust coefficient of the wind turbines to extract momentum at hub height. In doing so the model considers a modified surface stress due to the wind farm as well. Please update the paragraph (lines 31 to 37).

(9) The explaining text accompanying the figures in the results section (Section 3) is sometimes quite short.

(10) The chosen colour scale of several figures (especially Figs. 3 and 5) should be improved. It is extremely difficult to see the small differences which are said to be important.

References

Emeis, S., 2010: A simple analytical wind park model considering atmospheric stability. Wind Energy, 13, 459-469.

Sandu, I., A. Beljaars, P. Bechtold, T. Mauritsen, G. Balsamo, 2013: Why is it so difficult to represent stably stratified conditions in numerical weather prediction (NWP) models? J. Adv. Model. Earth Syst., 5, 117-133.

---

## Referee Comment (RC2) · Anonymous Referee #2 · 17 Jul 2017

The manuscript "Evaluation of the wind farm parametrization in the Weather Research and Forecast model (version 3.8.1) with meteorological and turbine data" compares the mesoscale model for different set-ups without wind farm parametrisation against Lidar data and the mesoscale model with and without a wind farm parametrisation against SCADA measurements.

The manuscript has a clear structure and is written well. The estimation of the wind farm power production is challenging and relevant to the wind energy community.

[Figure]

However, in my opinion the manuscript requires a more profound analysis before it can be published in GMD.

General comment

At least 4 uncertainties in simulating the power production with mesoscale models can be thought of, due to (I) mismatches in wind speed and wind direction (II) errors in simulating the wind speed reduction between grid-cells (III) errors in grid-cell internal wind speed reduction (IV) errors in power production, since turbine positions remain unresolved.

A big challenge in estimating the power production correctly is that the grid-cell averaged wind speed from the mesoscale model is not necessarily equivalent to the local up-stream wind speed of a turbine. Especially, in the presented wind farm layout, where turbines are systematically aligned in the West-East direction, mismatches between simulated and measured power production are expected. For southerly/northerly (and also south-westerly) winds, turbines in the southern/northern most rows all experience free-stream wind conditions and produce accordingly. On the other hand, the model wind speed in each turbine containing grid-cell is reduced due to the drag of all turbines in that grid-cell, leading to a systematically underestimated modelled power production. For easterly/westerly wind directions, the opposite happens and the interaction between turbines is underestimated, since in the model the turbine interaction is only area averaged.

Unfortunately, for the 4 day campaign not all wind directions are observed. However, there are examples that clearly show the challenges. On day 4 with south-westerly winds a large fraction of the turbines experience free-stream conditions, leading to large productions, whereas the modelled power production was systematically too

low. For events with southerly winds the situation is more complex, since only the front turbine rows experience free-stream conditions and all other turbines are affected by wind farm internal wakes.

Therefore, a deeper analysis on when and why the model is able/unable to estimate the power production is recommended. First, only events in which the wind speed and wind direction were modelled correctly should be included for the verification. Perhaps, in the scatter plots filled and empty circles could be used to distinguish events that were inside or outside the filter criteria.

One example: since Fig.9 shows that also for events with around 0 m/s bias the power production bias ranges from around -70 to 90 MW, it could be interesting to plot similar to Fig 8a the bias in power production against the modelled wind, but only for events where the wind speed bias was smaller than e.g. 2m/s and the wind direction bias smaller than e.g. 30°. In this plot the colours could be used to indicate the wind direction instead of the days. This would allow see the ability of simulating the power production for different wind directions.

For the above mentioned reasons, I would not agree with the conclusions made from the wind direction sensitivity analysis on p. 8 l. 228, given that the behaviour for the South-Westerly directions is very distinctive from other directions.

I agree that the model's performance is most likely not sensitive to the number of turbines per grid-cell, but is important to find out how sensitive it is to the turbine positions in the grid-cell.

Specific comments

p.2 l.57-59, WEP should be replaced with EWP.

p.4 l.114-123, in this paragraph measurements from the surface flux station are introduced. However, only the power bias against stability (Fig.8d) has been shown without showing the model's ability to simulate stability. For this figure to become more meaningful the model surface layer stability could be compared to that measured.

In the comparison of the TKE, it should be mentioned why the TKE derived from the Lidar measurements are expected to match the completely parametrised model TKE with a 1D production.

Fig. 7c, it is curious that the TKE of ERA12 is larger than that of ERA12WF as well in the night as during the day and that it seems never to be smaller. What would be your explanation for this?

Regarding section 2.2, here the model set-up (namelists) should probably be made available to GMD.

p.5 l.134, the inner domain with a 1 kilometer grid-spacing extends over the whole state of Iowa. This seems a computational expensive solution. What was the motivation for this configuration. Did you perform a sensitivity to the domain size?

In Fig. 3, it would be helpful to indicate the periods in which the 200S lidar and the WC lidar are in the wake of turbines.

---

## Author Response (AR1)

Responses to the reviewers are uploaded separately. Since the font and font size is fixed during the upload progress in GMD, the major and minor reviewer comments are numbered, with our responses indented.

REFEREE 1 COMMENTS:

We thank the reviewer for the thoughtful comments and suggestions and for their careful reading of our contribution.

Major comments:

(1) The study presented in this manuscript does not differentiate between deviations due to the simulation of the magnitude of the ambient wind speed and those due to inabilities of the wind farm parametrization (WFP). The WFP is not introduced in any detail and none of the found deviations is attributed to any feature of the WFP.

Thank you for the suggestion that we discuss the WFP in more detail. However, we are not introducing the WFP to the literature (as it has already been introduced by Fitch et al. (2012), as cited in our manuscript. Nevertheless, we now provide a brief summary of the WFP equations in section 2.2 for the interested reader.

The following is added to the manuscript, in lines 141 to 158:

"The WFP scheme simulates wind farms and their meteorological influences to the atmosphere. We provide a brief summary here, and the details are discussed in Fitch et al. (2012). Wind turbines slow down ambient wind flow and convert a part of the kinetic energy of wind into electrical energy. The WFP represents this wind-turbine drag force as the kinetic energy harvested by the turbine from the atmosphere:

$$F_{drag} = \frac{1}{2}C_T(|\boldsymbol{V}|)\rho|\boldsymbol{V}|A\boldsymbol{V},$$

where $C_T$ is the turbine-specific thrust coefficient (discussed in detail in Fitch, 2015), $\boldsymbol{V}$ is the horizontal velocity vector, $\rho$ is air density, $A = \frac{\pi}{4}D^2$ and is the cross-sectional rotor area, and $D$ is the rotor diameter. This kinetic-energy extraction also causes changes in the atmosphere, namely the kinetic energy loss in the grid cell, which is described by the momentum tendency:

$$\frac{\partial|\boldsymbol{V}|_{ijk}}{\partial t} = \frac{N_t^{ij}C_T(|\boldsymbol{V}|_{ijk})|\boldsymbol{V}|_{ijk}^2 A_{ijk}}{2(z_{k+1}-z_k)},$$

where $i$, $j$, and $k$ represents the zonal, meridional, and vertical grid indices, $N_t^{ij}$ is the number of wind turbines per square meter, and $z_k$ is the height at model level $k$. Of the kinetic energy extracted by the turbines, the WFP accounts for the electricity generation with:

$$\frac{\partial P_{ijk}}{\partial t} = \frac{N_t^{ij}C_P(|\boldsymbol{V}|_{ijk})|\boldsymbol{V}|_{ijk}^3 A_{ijk}}{2(z_{k+1}-z_k)},$$

where $P_{ijk}$ is the power output in the grid cell in Watts, and $C_P$ is the power coefficient. Assuming negligible mechanical and electrical losses, the rest of the kinetic energy harvested turns into TKE:

$$\frac{\partial TKE_{ijk}}{\partial t} = \frac{N_t^{ij}C_{TKE}(|\boldsymbol{V}|_{ijk})|\boldsymbol{V}|_{ijk}^3 A_{ijk}}{2(z_{k+1}-z_k)},$$

where $TKE_{ijk}$ is the TKE in the grid cell, and $C_{TKE}$ is the difference between $C_T$ and $C_P$."

Further, we did consider approaches for distinguishing between "deviations due to the simulation of the magnitude of the ambient wind speed and those due to inabilities of the WFP", but because there was no systematic bias (or even a stability-dependent bias) in the simulation of the ambient wind speed, such a differentiation is not possible. And in fact, performing and comparing idealized WRF simulations will address the implications of representing ambient wind speed accurately, while herein, we simulate a real case study since we aim to verify modelled power production with the observed. The focus here was on whether or not, as it stands, the WRF-WFP offers value beyond WRF itself, which had not yet been demonstrated in

past research. Attributing the deviations to the features of the WFP would require modifying those features, which would be the subsequent step in this research.

(2) Mesoscale models such as WRF are known to underrepresent the nocturnal low-level jet phenomenon. This has been analysed and explained by Sandu et al. (2013). Deviations in simulated power due to deviations in simulating the ramp effects at the onset of low-level jets (LLJ) have to be attributed to WRF itself and not to WFP. With regard to this known feature it seems a bit unlucky to
60    choose a LLJ episode for this WFP evaluation.

Thank you for the suggestion to include Sandu et al. (2013) in our literature review. Their analysis of the model performance of the 2011 version of the ECMWF model has some similarities with the current WRF framework. They note that "The ECMWF model operational
65    in 2011 reasonably represents this diurnal cycle of wind but underestimates its amplitude both at the surface and at 200 m." This type of behaviour is focused on how the ECMWF model predicts annual averages at the Cabauw tower, and is not borne out in WRF simulations in the Midwestern United States as well as Hamburg, Germany, as studied by Jahn et al. (2017), where they show some under-predictions as well as over-predictions, due to reductions in the stable
70    layer surface mixing in the MYNN scheme used herein.

Rather than being "unlucky" in our selection of cases, we specifically selected this case study because of the importance of LLJs to wind energy. Vanderwende et al. (2015) have also simulated the LLJ phenomenon for this case study. They found the WRF model overall is skilful
75    in simulating these LLJ events, as also demonstrated by our simulations. Moreover, LLJ events provide substantial wind resources, and verifying the WRF WFP in this period with ample power production is appropriate.

Although Sandu et al. (2013) tested the impact of turbulence diffusion and turbulence closure
80    using the ECMWF model, we do specifically acknowledge the potential influence of turbulence

parameterization in the revised version of the paper. Lines 345 to 347 now read "Reducing turbulence diffusion in the WRF model could potentially yield more accurate simulated winds in stable conditions, including LLJs (Sandu et al., 2013); active research in modifying mixing lengths (Jahn et al., 2017) is suggesting promising results."

85

(3) The lower right frame of Fig. 8 shows the dependence of the bias of the simulated power output from atmospheric stability. The authors interpret this figure as showing no significant dependence. My impression is, if the very few data points beyond the stability of 0.6 are skipped, that there is a significant influence of atmospheric stability (leading to a negative bias for more stable situations).

90

This is an interesting suggestion. If we remove the strong stability points ($z L^{-1}$ larger than 0.6), a weak negative correlation of -0.61 emerges. The dependence of negative power bias and stable atmosphere does increase when only data of accurately simulated WS and WD are considered (Fig 8d and the new Fig. 9d).

95

Therefore, we now state in lines 283 to 286 that "Moreover, when considering only cases of accurate wind predictions, the correlation between power bias and stability increases from -0.06 (Fig. 8d) to -0.42 (Fig. 9d). If the few strongest stability points ($z L^{-1}$ larger than 0.55) are removed from the analysis, a weak negative correlation with stability emerges as the Pearson correlation coefficient becomes -0.61."

100

(4) The discussion section (Section 4) makes reference to several results which have not been shown in the preceding results section (Section 3). Therefore, the reader cannot prove these conclusions.

105        Thank you for your comment. An introduction to the 2-sample t-test and the results from Table 6, 7 and 8 are now discussed in the results section. Lines 237 to 246 now read "Moreover, to statistically differentiate the power productions from various model runs, we apply the 2-sample Student's t-test. The null hypothesis of a 2-sample t-test is that the two population means are the

same, assuming the underlying distributions are Gaussian (Wilks, 2011). Hence, if the resultant
p-value is equal to or below 0.05, the two distributions are statically significantly different at the
95% confidence level. For example, the difference between the 4-day power-production
averages from the ERA12 and from the ERA12WF is -46.8 MW. The respective p-value is 0,
thus the difference of the means is statistically significant (Table 6). In other words, the ERA12
and the ERA12WF yield different power production distributions. Similarly, the GFS12 and the
GFS12WF lead to statistically different power outputs as the p-value from t-test is 0 as well
(Table 7). We also use the 2-sample t-test to contrast the actual and the modelled power
distributions. For instance, all the p-values between the no-WFP runs and the observation are 0,
implying those simulations yield power distributions significantly different from the reality
(Table 8)."

(5) The main conclusion that simulations with WFP are better than those without it is quite trivial.

In addition to our suggestion to use the WFP, we quantify the power-error reduction with the use
of the WFP, and we verify the WFP power by demonstrating the model power bias under
different circumstances including ambient wind speed and turbulence. We address questions of
performance such as the number of turbines per cell, etc. to help diagnose potential issues with
the WFP. We construct the verifications based on the usefulness of the WFP, which is one of the
key elements of this study.

(6) The study does not present any points which would allow for an enhancement of the simulation tool
(either WRF or WFP or both). (Please refer to comment (1) above as well)

We suggest that improving the wind speed prediction in the WRF model is the key to improve
the skills of the WFP. Please see lines 347 to 349, "More importantly, improving the skills of the
WRF model in simulating WS can improve the WFP power performance (Fig. 10). Future

versions of the WRF model as well as the WFP should aim to better account for instantaneous horizontal WS variations and the subsequent sub-gird wake interactions."

(7) sigma_u and sigma_w cannot be derived from a single monostatic remote sensing device (be it a sodar or a lidar) operating in Doppler beam swinging mode. Thus, the variables TKE and TI (equations (1) and (2)) are very unsure and cannot be used for a reliable evaluation study. This problem can be easily seen from the massive scatter of the TKE values in the third frame of Fig. 7.

The TKE and TI in equations (1) and (2) are calculated using 2-minute averaged variances of the 3-dimensional wind components measured by the lidar, as stated in the manuscript (lines 254 to 256) and as discussed in previous work by our group and others (Kumer et al., 2016; Rhodes and Lundquist, 2013). The trends of the modelled and observed TKE's shown in Fig. 7c match well, although the methodologies behind the two TKE's are spatially, temporarily and mathematically different.

To emphasize the differences between the TKE produced by the WRF model and the TKE derived from the lidar, lines 254 to 257 now read "Note that in Fig. 7c, the lidar derives TKE using 2-min variances, which is intrinsically different from the modelled TKE, as discussed in Rhodes and Lundquist (2013) and Kumer et al. (2016). Hence, readers should focus on the general trends of the TKE time series, rather than their absolute values."

Minor comments:

(8) The analytic wind farm model in Emeis (2010) is not based on an exaggerated surface roughness. This model uses the farm-averaged thrust coefficient of the wind turbines to extract momentum at hub height. In doing so the model considers a modified surface stress due to the wind farm as well. Please update the paragraph (lines 31 to 37).

Thank you for the clarification. After revisiting Emeis (2010), we agree that the effective roughness length at hub height depends on the effective drag coefficient, which is a sum of both the areal average of momentum extraction coefficient of turbines (or thrust coefficient, as mentioned in your comment) and the surface drag coefficient.

Lines 34 to 37 now read "Similarly, the analytical wind park model of Emeis and Frandsen (1993) considers both the downward momentum flux and the momentum loss due to surface roughness. The revised model by Emeis (2010) accounts for the spatially-averaged momentum-extraction coefficient by turbines, and the parameters become atmospheric-stability dependent."

(9) The explaining text accompanying the figures in the results section (Section 3) is sometimes quite short.

The results section now includes the following text:

Lines 191 to 195 now read "The 200S records the vertical shear caused by LLJs above 100 m (Fig. 3a), and the WC measures the near-surface winds with high temporal resolution (Fig. 3b). In both the observations and the simulations of WS (Fig. 3c), the night-time WS profile is stratified and the daytime atmosphere is well-mixed. The WD simulations also match well with the measurements, where in the evening the winds veer, or turn clockwise with height (Fig. 4), while the daytime flow shows relatively constant wind direction with height."

Lines 214 to 216 now read "We present the total 10-min observed and simulated power of the whole wind farm at the bottom of each panel in Fig. 5, where the total power production of the WFP run matches the observed."

Based on the new Fig. 9, lines 279 to 287 now read "To isolate the WFP errors in power predictions from the WRF model errors in ambient wind simulations, we analyse a subset of

data where the winds are simulated accurately. When the absolute error in wind speed is smaller than 1 m s$^{-1}$ and the absolute error in wind direction is smaller than 5°, the relationships between power bias and WS, WD and TI (Fig. 9a to c) remain similar to the general trends shown in Fig. 8a to c. The WS-power-bias and TI-power-bias correlations become stronger in this subset (Fig 9a and c), compare with all the data in the 4-day period (Fig 8a and c). Moreover, when considering only cases of accurate wind predictions, the correlation between power bias and stability increases from -0.06 (Fig. 8d) to -0.42 (Fig. 9d). If the few strongest stability points (z L$^{-1}$ larger than 0.55) are removed from the analysis, a weak negative correlation with stability emerges as the Pearson correlation coefficient becomes -0.61. Additionally, generally south to south-westerly flows yield stronger negative power biases."

In lines 303 to 308, "Furthermore, the WFP performance remains consistent between upwind and downwind turbines, based on their positions against the ambient winds (Fig. 12). Given the square shape of grid cells, we determine the sequential rows of turbines during strictly southerly flows, with WD between 175° and 185° (Fig. 12a). The bulk of the normalized power biases fall within 0 to 0.4 MW, regardless of the upwind-downwind positions of turbines. Additionally, the power bias is independent of the mean distance between the actual turbine locations and the centre points of their respective grid cells (not shown)."

(10) The chosen colour scale of several figures (especially Figs. 3 and 5) should be improved. It is extremely difficult to see the small differences which are said to be important.

We have adjusted the dark end of the colour table for Figs. 3 and 5 to emphasize the range of values. Please see the revised figures below.

[Figure]

[Figure]

ERA12
Total = 234.0 MW

ERA12WF
Total = 186.6 MW

OBS
Total = 194.7 MW

220

REFEREE 2 COMMENTS:

We thank the reviewer for their careful reading of our manuscript and their comments and suggestions.

225

Major comments:

(1) At least 4 uncertainties in simulating the power production with mesoscale models can be thought of, due to (I) mismatches in wind speed and wind direction (II) errors in simulating the wind speed 230 reduction between grid-cells (III) errors in grid-cell internal wind speed reduction (IV) errors in power production, since turbine positions remain unresolved.

A big challenge in estimating the power production correctly is that the grid-cell averaged wind speed from the mesoscale model is not necessarily equivalent to the local up-stream wind speed of a turbine. 235 Especially, in the presented wind farm layout, where turbines are systematically aligned in the West-East direction, mismatches between simulated and measured power production are expected. For southerly/northerly (and also south-westerly) winds, turbines in the southern/northern most rows all experience free-stream wind conditions and produce accordingly. On the other hand, the model wind speed in each turbine containing grid-cell is reduced due to the drag of all turbines in that grid-cell, 240 leading to a systematically underestimated modelled power production. For easterly/westerly wind directions, the opposite happens and the interaction between turbines is underestimated, since in the model the turbine interaction is only area averaged.

Unfortunately, for the 4 day campaign not all wind directions are observed. However, there are 245 examples that clearly show the challenges. On day 4 with south-westerly winds a large fraction of the turbines experience free-stream conditions, leading to large productions, whereas the modelled power production was systematically too low. For events with southerly winds the situation is more complex,

since only the front turbine rows experience free-stream conditions and all other turbines are affected by wind farm internal wakes.

250

Therefore, a deeper analysis on when and why the model is able/unable to estimate the power production is recommended. First, only events in which the wind speed and wind direction were modelled correctly should be included for the verification. Perhaps, in the scatter plots filled and empty circles could be used to distinguish events that were inside or outside the filter criteria.

255

One example: since Fig.9 shows that also for events with around 0 m/s bias the power production bias ranges from around -70 to 90MW, it could be interesting to plot similar to Fig 8a the bias in power production against the modelled wind, but only for events where the wind speed bias was smaller than e.g. 2m/s and the wind direction bias smaller than e.g. 30°. In this plot the colours could be used to indicate the wind direction instead of the days. This would allow see the ability of simulating the power production for different wind directions.

For the above mentioned reasons, I would not agree with the conclusions made from the wind direction sensitivity analysis on p. 8 l. 228, given that the behaviour for the South-Westerly directions is very distinctive from other directions.

Thank you for the thoughtful suggestion. According to your suggestion, we produce the following plot as the new Figure 9. This figure replicates Fig. 8, while only the data with the modelled-observed absolute error in wind speed smaller than 1 m s$^{-1}$ and the absolute error in wind direction smaller than 5°. In other words, the WRF model reproduces accurate ambient winds in these data points. Different colours represent different lidar-measured wind directions and the n's are respective sample sizes.

[Figure]

275

In Fig 8d, although the correlation between power bias and stability is only -0.06 if we include all the data points, when we only consider data points when the WRF model simulates accurate wind speed and wind direction, the correlation increases to -0.42.

280

In the new Fig. 9, the general trend of the negative relationship between power bias and measured wind speed shown in Fig. 8a remains prominent. When wind speed is below 9 m s$^{-1}$, power bias is mostly above -30 MW, regardless of wind direction.

From the new Fig. 10 (the original Fig. 9), power bias overall is positively correlated with wind speed bias, such that overestimating wind speed leads to power overestimation. The new Fig. 9

285

shows that even without large errors in simulating wind speed and wind direction, strong winds (in this case, wind speed at around and beyond 10 m s$^{-1}$) actually lead to under-prediction of power. The power underestimation is associated with southerly to south-westerly winds, and the respective wind speeds are also relatively large. Hence we cannot confidently conclude the interactions between wind direction and wind-farm layout and their resultant influence on power production and wake effects as suggested by the reviewer.

One possible factor contributing to the consistent underestimation of power production during south-westerly flow is the wake interactions within a grid cell. However, we have already shown that inter-cell wake effects are not the critical factor to power error (the new Fig. 12b). The inability of the WFP to simulate intra-cell wake effects can explain the large negative biases when many of the turbines experience unobstructed south-westerly flow.

Based on the new figure, the following are added to the text. In lines 279 to 287 (the results section),

"To isolate the WFP errors in power predictions from the WRF model errors in ambient wind simulations, we analyse a subset of data where the winds are simulated accurately. When the absolute error in wind speed is smaller than 1 m s$^{-1}$ and the absolute error in wind direction is smaller than 5°, the relationships between power bias and WS, WD and TI (Fig. 9a to c) remain similar to the general trends shown in Fig. 8a to c. The WS-power-bias and TI-power-bias correlations become stronger in this subset (Fig 9a and c), compare with all the data in the 4-day period (Fig 8a and c). Moreover, when considering only cases of accurate wind predictions, the correlation between power bias and stability increases from -0.06 (Fig. 8d) to -0.42 (Fig. 9d). If the few strongest stability points (z L$^{-1}$ larger than 0.55) are removed from the analysis, a weak negative correlation with stability emerges as the Pearson correlation coefficient becomes -0.61. Additionally, generally south to south-westerly flows yield stronger negative power biases."

In lines 350 to 358 (the discussion section),

"Besides necessary improvements in simulating ambient WS, the WFP scheme itself also requires refinements. When background winds are accurately predicted, the power-bias dependence on WS and TI remain strong (Fig. 9a and c). Although the relationship between the WFP performance and stability is generally indistinct, the correlation becomes weakly negative without the strongly stable data. Even when the simulated winds are close to observations, the WFP underestimates power during high WS, south to south-westerly flow, low TI and stable conditions. Certainly the interactions between WD and wind-farm layout affect the power-bias relationships, while further sensitivity tests can further upgrade the WFP performance, particularly in intra-cell WS reduction. We demonstrate that inter-cell wake effects are not the critical factor to power error (Fig. 12b), hence the inability of the WFP to simulate intra-cell wake effects can explain the biases when many of the turbines experience accurately-simulated ambient flow."

(2) I agree that the model's performance is most likely not sensitive to the number of turbines per grid-cell, but is important to find out how sensitive it is to the turbine positions in the grid-cell.

Following the suggestion, we have explored different ways of considering the turbine positions in the grid-cell, and we found no sensitivity. Below we first show the mean power bias in each turbine-containing grid cell of the ERA12WF simulation as a function of the mean distance between the actual turbine locations and the centre point of their respective grid cells, within a grid cell.

[Figure]

340    In this second plot below, we look at only the north-south distance within the cell, between the
actual turbine location and the grid cell centre. The x-axis below is the sum of distances between
turbine latitude and grid centre latitude, based on the longitude of the grid cell centre point.
Positive values indicate the wind turbine location is south of the grid centre, or grid centre
latitude is larger than the turbine latitude. Negative values indicate the wind turbine location is
345    north of the grid centre. The 4-day correlation is 0.23.

[Figure]

When the turbine locations are farther from the centre of a grid cell, the power bias stays relatively consistent around zero. The lack of a distinct relationship between power bias and observed-modelled distance demonstrates that the turbine position in grid cell does not have a significant role in power error. This is reasonable since the mesoscale WRF model indicates turbines and their effects across the whole grid cell, as long as they are within that grid cell.

These results are added to the text, in lines 306 to 308, "Additionally, the power bias is independent of the mean distance between the actual turbine locations and the centre points of their respective grid cells (not shown)."

Minor comments:

(3) p.2 l.57-59, WEP should be replaced with EWP.

All the "WEP" are now corrected to "EWP". Thank you.

(4) p.4 l.114-123, in this paragraph measurements from the surface flux station are introduced.
However, only the power bias against stability (Fig.8d) has been shown without showing the model's
ability to simulate stability. For this figure to become more meaningful the model surface layer stability
could be compared to that measured.

Since stability parameters like Obukhov length are not the standard output variables of the WRF
model, we use the sensible heat flux to represent stability in the revised Fig. 7d.

Lines 252 to 254 now read, "Although the magnitudes of the surface sensible heat flux of the
surface flux station and the simulations differ, their signs change at similar times, particularly in
the last three days (Fig. 77d). Hence the WRF model is capable to represent diurnal atmospheric
stability changes."

[Figure]

(5) In the comparison of the TKE, it should be mentioned why the TKE derived from the Lidar measurements are expected to match the completely parametrised model TKE with a 1D production.

Similar to our answer to comment (7) of Referee #1, only the general trends of lidar TKE and model TKE should match, rather than their values. Lines 254 to 257 now read "Note that in Fig. 7c, the lidar derives TKE using 2-min variances, which is intrinsically different from the modelled TKE, as discussed in Kumer et al. (2016) and Rhodes and Lundquist (2013). Hence,

readers should focus on the general trends of the TKE time series, rather than their absolute values."

(6) Fig. 7c, it is curious that the TKE of ERA12 is larger than that of ERA12WF as well in the night as during the day and that it seems never to be smaller. What would be your explanation for this?

Most of the TKE deviations between the two simulations take place before 12 UTC on 24 August, where the ERA12 simulates higher wind speeds than the ERA12WF. The reduced wind speeds in the ERA12WF simulation can explain lowered TKE in the evening, as the vertical momentum transfer is obstructed.

(7) Regarding section 2.2, here the model set-up (namelists) should probably be made available to GMD.

An example namelist is now uploaded to http://doi.org/10.5281/zenodo.847780 and the link provided in the revised version of the manuscript.

(8) p.5 l.134, the inner domain with a 1 kilometre grid-spacing extends over the whole state of Iowa. This seems a computational expensive solution. What was the motivation for this configuration. Did you perform a sensitivity to the domain size?

We include the whole state of Iowa in the smallest domain in order to ensure the wind farm is near the centre of the domain, and not to be close to the domain edges. Since the wind direction changes over the four-day period, locating the wind site near the domain centre is an appropriate choice, even the domain can be slightly smaller. We did not perform any sensitivity study with respect to the size of the domain. Moreover, simulating the whole state of Iowa also provides model data for future meteorological comparisons in other sites in western Iowa with wind farm

data that our group is in the process of acquiring (as in Walton et al., 2014), so the model simulation output will be used for other purposes as well.

415

(9) In Fig. 3, it would be helpful to indicate the periods in which the 200S lidar and the WC lidar are in the wake of turbines.

The wind direction changes from southeast to southwest during the 4-day period. The WC lidar
420    measures ambient flow. The nearest upwind turbine, southeast of the WC lidar, is located over 32 D (2.7 km) away. Hence the WC observations are only potentially affected by wakes when the winds are south-easterly, which is during the first 30 hours of the case study. And at that distance, it would be remarkable for a wake to affect the WC.

425    The 200S lidar is 440 m north of a turbine row, so its measurements in the lowest 120 m of the atmosphere may have been waked on occasion. However, the 200S measurements used herein (Figs. 3 and 4) are primarily above the top of the turbine rotor disk (120 m) and would not be affected by wakes. We have now included a reference to a manuscript in which the 200S horizontal scans are used to understand turbine wake behaviour in this wind farm (Bodini et al.,
430    2017).

We think adding the above as graphic indications would overcrowd the contour plots (Fig. 3 and Fig. 4), so we include these notes in the text. Lines 113 to 117 now read "Since the dominant wind directions during the campaign are south-easterly to south-westerly (Vanderwende et al.,
435    2015), some of the 200S measurements below the rotor top (about 120 m AGL) could be influenced by turbine wakes during conditions, in which wakes persist longer than 5 D downwind from the turbine (Bodini et al., 2017). On the other hand, WC measurements are largely unaffected by turbine wakes except when WD is east of 150°. The closest upwind turbine during this simulation period was located over 2.7 km (33 D) to the southeast."
440    Moreover, lines 196 to 198 now read "The 200S measurements above the rotor layer (120 m) are

[revised manuscript text omitted]

---

## Referee Report (RR1)

The comments have generally be addressed very satisfactory. Only on the reply to the first general comment, I would like to comment. Below, only the open comments are listed (authors' reply is in italic and new comments in normal serif). I used the line numbers from the review report.

*l. 265: Thank you for the thoughtful suggestion. According to your suggestion, we produce the following plot as the new Figure 9. This figure replicates Fig. 8, while only the data with the modelled-observed absolute error in wind speed smaller than $1\,m\,s^{-1}$ and the absolute error in wind direction smaller than $5°$. In other words, the WRF model reproduces accurate ambient winds in these data points. Different colours represent different lidar-measured wind directions and the ns are respective sample sizes.*
From the new figure 9, it seems to me that stability and wind direction could be both key parameters.

*l. 280: In Fig 8d, although the correlation between power bias and stability is only -0.06 if we include all the data points, when we only consider data points when the WRF model simulates accurate wind speed and wind direction, the correlation increases to -0.42. In the new Fig. 9, the general trend of the negative relationship between power bias and measured wind speed shown in Fig. 8a remains prominent. When wind speed is below $9\,m\,s^{-1}$, power bias is mostly above -30 MW, regardless of wind direction.* The power depends strongly on the wind speed, therefore the power bias in function of the wind speed can be expected to increase with wind speed. To me, it seems that the positive power bias for wind speeds up-to 8-9 $m\,s^{-1}$ is mainly mainly caused by the unstable atmospheric conditions.

*l. 290: From the new Fig. 10 (the original Fig. 9), power bias overall is positively correlated with wind speed bias, such that overestimating wind speed leads to power overestimation. The new Fig. 9 shows that even without large errors in simulating wind speed and wind direction, strong winds (in this case, wind speed at around and beyond $10\,m\,s^{-1}$) actually lead to under-prediction of power. The power underestimation is associated with southerly to south-westerly winds, and the respective wind speeds are also relatively large. Hence we cannot confidently conclude the interactions between wind direction and wind-farm layout and their resultant influence on power production and wake effects as suggested by the reviewer.*
The increasing negative power bias for higher wind speeds in stable conditions is probably the result of the wind's dependency on the power. To me figure 9d shows that the model generally overestimates the power generation in unstable/near neutral conditions (underestimates the wake losses) and underestimates the power generation in stable conditions (wake losses are too high). Figure 9b, shows that for 160° (mostly stable) the bias is around 0%. For southerly wind directions the bias becomes negative in stable cases. Then, for south-westerly winds it goes only back to around 0% only because the atmospheric conditions changed to unstable/neutral. Therefore, for similar atmospheric stability there

is a sensitivity to the wind direction. Namely that for southerly winds where wake losses in reality are expected to be low (since the turbine spacing is larger then with e.g. 270°) the model overestimates the wake losses. What is your opinion?

*l 295: One possible factor contributing to the consistent underestimation of power production during south-westerly flow is the wake interactions within a grid cell. However, we have already shown that inter-cell wake effects are not the critical factor to power error (the new Fig. 12b). The inability of the WFP to simulate intra-cell wake effects can explain the large negative biases when many of the turbines experience unobstructed south-westerly flow.*

Couldn't figure Fig. 12b indicate that for southerly winds the first row systematically underestimates the power production, which can only be due to grid-cell internal wind speed reductions. Further in the wind farm, it becomes more complicated, since there are grid-cell internal wind speed reductions and reductions caused by upstream turbines. The fluctuations in power bias in the following rows could then be caused by wind speed reductions from upstream turbines. Unfortunately, there is no data for 90 or 270°, but for those directions the opposite could hold. In case the power bias in Fig. 12b is calculated in the same way as in the previous plots, shouldn't the vertical axis be reversed (since the bias was previously mostly negative).

*l 335: Following the suggestion, we have explored different ways of considering the turbine positions in the grid-cell, and we found no sensitivity. Below we first show the mean power bias in each turbine-containing grid cell of the ERA12WF simulation as a function of the mean distance between the actual turbine locations and the centre point of their respective grid cells, within a grid cell.*

My comment was more related to the previous discussion that in reality it can happen that there is no (180°) or a very intensive (270°) interaction between turbines in one grid-cell, whereas the wind speed reduction in the model would be exactly the same. Since the wind farm operates in complex conditions and the data amount is limited it is not straight to assess this.

---

## Author Response (AR2)

Responses to the reviewers are uploaded separately. Since the font and font size is fixed during the upload progress in GMD, the major and minor reviewer comments are numbered, with our responses indented.

REFEREE 2 COMMENTS:

> We thank the reviewer for their careful reading of our revised manuscript and their comments and suggestions. We think comment (1), (2) and (3) are connected, hence we reply to those comments altogether.

(1) From the new Figure 9, it seems to me that stability and wind direction could be both key parameters.

(2) The power depends strongly on the wind speed, therefore the power bias in function of the wind speed can be expected to increase with wind speed. To me, it seems that the positive power bias for wind speeds up-to 8-9 m s$^{-1}$ is mainly caused by the unstable atmospheric conditions.

(3) The increasing negative power bias for higher wind speeds in stable conditions is probably the result of the wind's dependency on the power. To me Figure 9d shows that the model generally overestimates the power generation in unstable/near neutral conditions (underestimates the wake losses) and underestimates the power generation in stable conditions (wake losses are too high). Figure 9b, shows that for 160° (mostly stable) the bias is around 0%. For southerly wind directions the bias becomes negative in stable cases. Then, for south-westerly winds it goes only back to around 0% only because the atmospheric conditions changed to unstable/neutral. Therefore, for similar atmospheric stability there is a sensitivity to the wind direction. Namely that for southerly winds where wake losses in reality are expected to be low (since the turbine spacing is larger than with e.g. 270°) the model overestimates the wake losses. What is your opinion?

> According to your previous suggestions, we found changes in wind direction and stability influence power bias, due to the shape of the wind farm. Both variables interact with wind speed changes over the 4-day period as well, as the meteorological changes are bundled together in this real-case simulation study.

> The following is a modification of Fig. 9, where the filled dots represent the unstable data points ($z L^{-1}$ smaller than 0), and the hollow circles represent the stable data points ($z L^{-1}$ larger than 0). We replaced the current Fig. 9 with the following.

[Figure]

Figure 1: As in Fig. 9, and only including data when the winds are accurately simulated in the ERA12WF run: the modelled-observed absolute error in WS smaller than 1 m s$^{-1}$ and the absolute error in WD smaller than 5°. Different colours represent different WD bins: 150° to 170° in blue, 170° to 190° in cyan, 190° to 210° in orange, 210° to 230° in red, and 230° and beyond in maroon. The n values illustrate the respective sample size in each wind-direction bin. Solid circles represent unstable conditions (z L$^{-1}$ smaller than 0) and hollow circles represent stable conditions (z L$^{-1}$ larger than 0).

From the above figure, positive biases, including those with wind speed within 8 to 9 m s$^{-1}$, occur in both stable and unstable conditions, different from the opinion of the reviewer.

Most of the unstable cases are associated with positive power biases. The judgement of the reviewer is correct, in which power biases are negative when winds change from south-easterly

to south-westerly during stable conditions. However, one should be aware that there are only 27 unstable cases and 125 stable cases shown in Fig. 9.

55    Lines 283 to 290 now read, "Moreover, when considering only cases of accurate wind predictions, the correlation between power bias and stability increases from -0.06 (Fig. 9d) to -0.42 (Fig. 9d). In the few (27) unstable conditions with accurate wind speed predictions, the power bias is generally positive, given moderate WS and high TI (Fig 9 a, c and d). In the stable regime, the WFP tends to underestimate power, regardless of WD (Fig. 9 b and d): 106 of the

60    125 stable data points are under-predicted. If the few strongest stability points ($z L^{-1}$ larger than 0.55) are removed from Fig. 9d, a weak negative correlation with stability emerges as the Pearson correlation coefficient becomes -0.61. Additionally, generally south to south-westerly flows yield stronger negative power biases."

65    Lines 353 to 361 now read, "When background winds are accurately predicted, the power-bias dependence on WS and TI remain strong (Fig. 10a and c). The correlation between the WFP performance and atmospheric stability becomes weakly negative without the strongly stable data (Fig. 9d). Even when the simulated winds are close to observations, the WFP tends to underestimate power during high WS, low TI and stable conditions. In contrast, the WFP tends

70    to over-predict power in unstable, turbulent conditions, with the caveat that a small number of unstable cases are considered here. The WFP scheme appears to overestimate wake loss within a grid cell in stable and windy conditions, and underestimate wake effects in an unstable and well-mixed atmosphere. Certainly the interactions between WD and wind-farm layout affect the power-bias relationships, while further sensitivity tests can provide more insight into the WFP

75    performance, particularly in intra-cell WS reduction."

(4) Couldn't Figure Fig. 12b indicate that for southerly winds the first row systematically underestimates the power production, which can only be due to grid-cell internal wind speed reductions. Further in the wind farm, it becomes more complicated, since there are grid-cell internal wind speed reductions and

80    reductions caused by upstream turbines. The fluctuations in power bias in the following rows could then be caused by wind speed reductions from upstream turbines. Unfortunately, there is no data for 90° or 270°, but for those directions the opposite could hold. In case the power bias in Fig. 12b is calculated in the same way as in the previous plots, shouldn't the vertical axis be reversed (since the bias was previously mostly negative).

85

Fig. 12b illustrates data points from strictly southerly WRF-simulated flows (according to the grid cell closest to the WC lidar), which is different from the WC-recorded wind directions shown in Fig. 8b and Fig. 9b. The ERA12WF run generally yields positive power biases during

those simulated southerly cases, eg. around 17Z 24 August, agreeing the power over-estimation
90    shown in Fig. 12b. The message from Fig. 12 is to demonstrate simulated inter-cell wakes does
not affect power bias, hence we selected data of simulated southerly flows, instead of observed
southerly flows.

(5) My comment was more related to the previous discussion that in reality it can happen that there is no
95  (180°) or a very intensive (270°) interaction between turbines in one grid-cell, whereas the wind speed
reduction in the model would be exactly the same. Since the wind farm operates in complex conditions
and the data amount is limited it is not straight to assess this.

Thank you for your clarification and thank you for your understanding. We hope to explore this
100    in the future when we have more data available from wind farms with different geometry.

[revised manuscript text omitted]
 | 178.7                |                     | 7.8                 | 0.0136     |